# Radar-based characterisation of heavy precipitation in the eastern Mediterranean and its representation in a convection-permitting model

**Moshe Armon**[1]**, Francesco Marra**[1,2]**, Yehouda Enzel**[1]**, Dorita Rostkier-Edelstein**[1,3]**, and Efrat Morin**[1]

[1]Fredy and Nadine Herrmann Institute of Earth Sciences, the Hebrew University of Jerusalem, Edmond J. Safra Campus, Jerusalem 9190401, Israel
[2]National Research Council of Italy, Institute of Atmospheric Sciences and Climate, CNR-ISAC, Bologna 40129, Italy
[3]Department of Applied Mathematics, Environmental Sciences Division, IIBR, Ness-Ziona 7410001, Israel

**Correspondence:** Moshe Armon (moshe.armon@mail.huji.ac.il)

**Abstract.** Heavy precipitation events (HPEs) can lead to natural hazards (floods, debris flows) and contribute to water resources. Spatiotemporal rainfall patterns govern the hydrological, geomorphological and societal effects of HPEs. Thus, a correct characterisation and prediction of rainfall patterns is crucial for coping with these events. Information from rain gauges is generally limited due to the sparseness of the networks, especially in the presence of sharp climatic gradients. Forecasting HPEs depends on the ability of weather models to generate credible rainfall patterns. This paper characterises rainfall patterns during HPEs based on high-resolution weather radar data and evaluates the performance of a high-resolution, convection-permitting Weather Research and Forecasting (WRF) model in simulating these patterns. We identified 41 HPEs in the eastern Mediterranean from a 24-year radar record using local thresholds based on quantiles for different durations, classified these events into two synoptic systems, and ran model simulations for them. For most durations, HPEs near the coastline were characterised by the highest rain intensities; however, for short durations, the highest rain intensities were characterised for the inland desert. During the rainy season, the rain field's centre of mass progresses from the sea inland. Rainfall during HPEs is highly localised in both space (<10 km decorrelation distance) and time (<5 min). WRF model simulations were accurate in generating the structure and location of the rain fields in 39 out of 41 HPEs. However, they showed a positive bias relative to the radar estimates and exhibited errors in the spatial location of the heaviest precipitation. Our results indicate that convection-permitting model outputs can provide reliable climatological analyses of heavy precipitation patterns; conversely, flood forecasting requires the use of ensemble simulations to overcome the spatial location errors.

## 1 Introduction

Heavy precipitation events (HPEs) cause natural hazards such as flash, riverine, and urban floods, landslides and debris flows; they also serve as a resource for recharging ground- and surface-water reservoirs (e.g., Bogaard and Greco, 2016; Borga et al., 2014; Borga and Morin, 2014; Doswell et al., 1996; Nasta et al., 2018; Raveh-Rubin and Wernli, 2015; Samuels et al., 2009; Taylor et al., 2013; UN-Habitat, 2011). Diverse rainfall patterns during HPEs cause different hydrological responses, thus an accurate representation of rainfall patterns during these events is crucial to detecting and predicting climate change-induced precipitation changes (Maraun et al., 2010; Trenberth et al., 2003). In particular, understanding the specific interactions between rainstorms and catchments is critical in small watersheds, where accurate, high spatiotemporal resolution observations and forecasts are required (e.g., Bloschl and Sivapalan, 1995; Cristiano et al., 2017). However, these data may not be available through operational tools, such as rain gauge networks and coarse-scale weather models (e.g., commonly used, global or even regional circulation models). High-resolution observation and HPE forecasts thus remain a challenge (Borga et al., 2011; Collier, 2007; Doswell et al., 1996).

Rain gauge data can be used to quantify general characteristics of HPEs (such as rain intensity and depth on a point

scale), but their density is generally insufficient to adequately represent the spatial gradients, particularly in the case of sparsely gauged regions, short-span events, and arid climates (Amponsah et al., 2018; Kidd et al., 2017; Morin et al., 2009, 2019). This problem is enhanced in regions characterised by high climatic gradients such as the eastern Mediterranean (EM) (El-Samra et al., 2017; Marra et al., 2017; Marra and Morin, 2015; Morin et al., 2007; Rostkier-Edelstein et al., 2014). Thus, a characterisation of HPEs with high resolution in such regions must be supported by other types of records. Remotely sensed precipitation estimates, such as those acquired from weather radars, provide the necessary spatiotemporal resolutions (e.g., 1 km, 5 min) and coverage (regional scale), and have been shown to be useful for analysing specific events (e.g., Borga et al., 2007; Dayan et al., 2001; Krichak et al., 2000; Smith et al., 2001). Where continuous radar records exist, they have been used in climatological studies as well (Belachsen et al., 2017; Bližňák et al., 2018; Peleg and Morin, 2012; Saltikoff et al., 2019; Smith et al., 2012). However, climatological characterisations of rainfall patterns during HPEs are rare in the literature and often based on rain gauge identification of those events (Panziera et al., 2018; Thorndahl et al., 2014).

High-resolution numerical weather prediction (NWP) models allow simulating and forecasting HPEs, and as added value, enable understanding their past and present patterns, and a prediction of possible future behaviours (Cassola et al., 2015; Deng et al., 2015; El-Samra et al., 2017; Kendon et al., 2014; Prein et al., 2015; Rostkier-Edelstein et al., 2014; Yang et al., 2014). In particular, convection-permitting models are increasingly used in weather forecasts, climatological studies and event-based reanalyses (e.g., Ban et al., 2014; Fosser et al., 2014; Hahmann et al., 2010; Khodayar et al., 2016; Prein et al., 2015; Rostkier-edelstein et al., 2015). Such models downscale global or regional NWP models, and provide a direct representation of convective rainfall that, due to its high intensity and local characteristics, often plays a major role in HPEs (e.g., Flaounas et al., 2018). In addition, these models can provide 3-D fields of otherwise unmeasurable meteorological variables, thus contributing to our understanding of the dynamics of HPEs. Studies based on high-resolution NWP models commonly focus on specific cases. For example, Zittis et al. (2017) examined the performance of a high-resolution NWP model during five HPEs in the EM, and identified large discrepancies between grid- and gauge-based precipitation datasets, making it hard to validate the model. Only a few studies have examined the climatology of model results, to either determine the atmospheric conditions that trigger HPEs, or understand the overall rainfall pattern in comparison to observational records (e.g., Flaounas et al., 2019; Kendon et al., 2014; Khodayar et al., 2018). Commonly, climate change studies based on high-resolution NWP models characterise the expected changes in precipitation, focusing on rainfall intensity or frequency, or some

derived index (e.g., Ban et al., 2014; Hochman et al., 2018b; Schär et al., 2016; Westra et al., 2014).

A basic question, however, remains open: To what degree is the model description of rainfall during HPEs credible? Moreover, the model's ability to reproduce rainfall patterns can differ among synoptic types. To answer this question, both a realistic spatiotemporal representation of rainfall during HPEs and a large number of observed HPEs, triggered by various synoptic systems, are necessary. In this paper, we present a successful step in this direction based on a corrected and calibrated 24-year-long record of weather radar data recently developed for the EM, and found to adequately represent extreme precipitation events (Marra and Morin, 2015). As an essential step in understanding and quantifying rainfall-generating processes involved in HPEs, and as a basis for a future study that will include downscaling of climate change projections to understand changes in rainfall patterns, here we aim to (i) characterise high-resolution rainfall patterns (seasonality, spatial distribution of intensities, location, and spatiotemporal structure) during HPEs in the hydroclimatically heterogeneous EM, and (ii) assess the capabilities of a regional convection-permitting weather model to simulate these patterns. To this aim, we identified all HPEs embedded in the radar record (41 events), and simulated them using a convection-permitting Weather Research and Forecasting (WRF) model (Skamarock et al., 2008). This long and consistent high-resolution dataset is unique, and is therefore interesting both for examining HPE climatology, and as a basis for convection-permitting model evaluation. Considering that our observations are based on radar data, they are certainly not perfect. Therefore, we quantified and compared several rainfall characteristics from both radar estimates and simulated rainfall to evaluate the model's ability to reproduce the rainfall patterns and to obtain climatological characteristics of HPEs.

The paper is structured as follows: Section 2 describes the study region. The radar and weather model data are explained in Sect. 3.1 and 3.2, respectively. Identification and synoptic classification of HPEs are presented in Sect. 3.3 and 3.4, respectively. The methods used in evaluating model performance are presented in Sect. 3.5. Section 4 presents the results of the evaluation and characterisation of rainfall patterns during HPEs. Section 5 provides a discussion and Sect. 6 concludes.

## 2   Study region

This study focuses on the EM region, where Mediterranean climate (with parts of it receiving mean annual precipitation $>1000$ mm year$^{-1}$) drops to hyperarid ($<50$ mm year$^{-1}$) over a short distance (Goldreich, 2012) (Fig. 1). Precipitation is dominated by rainfall, and occurs mainly between October and May, with summer months (June–September) being essentially dry (Kushnir et al., 2017). Most of this rainfall

is associated with cold north-westerly flows in the rear part of Mediterranean Cyclones (MCs). These MCs pass above the warm water of the Mediterranean Sea, absorbing moisture and precipitating it over the EM region (Alpert et al., 2004; Alpert and Shay-EL, 1994; Armon et al., 2019; Saaroni et al., 2010; Ziv et al., 2015). High surface water temperature favours high-intensity rainfall and floods, most commonly at the beginning of the rainy season and near the sea. As the MCs move inland and towards the desert, a substantial amount of the moisture is lost, and rainfall occurrence and amounts are greatly reduced (Enzel et al., 2008). In this arid region, HPEs are associated not only with MCs (Kahana et al., 2002), but also with Active Red Sea Troughs (ARSTs) (Ashbel, 1938; Krichak et al., 1997; De Vries et al., 2013) and, more rarely, with Tropical Plumes (Armon et al., 2018; Rubin et al., 2007; Tubi et al., 2017). Commonly, rainfall during ARSTs is of a spotty nature, can reach far into the desert, and can be of very high intensity (Armon et al., 2018; Sharon, 1972). Conversely, during Tropical Plumes, rainfall is widespread, potentially covering most of the region simultaneously with moderate intensities. Desert HPEs frequently result in large and sometimes devastating flash floods (e.g., Armon et al., 2018; Dayan and Morin, 2006; Farhan and Anbar, 2014; Kahana et al., 2002; Saaroni et al., 2014; Seager et al., 2014). Projections for precipitation in the EM indicate a substantial decrease in annual rainfall amounts (Giorgi and Lionello, 2008); however, the importance of credible HPE simulations stems from, among others, opposing trends that may appear between number and intensity of HPEs generated by different synoptic conditions (Alpert et al., 2002; Hochman et al., 2018b, 2019; Marra et al., 2019); for example, based on Dead Sea sedimentological data, it has been suggested that when MC frequency is reduced, i.e., there is a regional drought, the frequency of HPEs generated by ARSTs may increase (Ahlborn et al., 2018).

## 3 Methodology and data

### 3.1 Weather radar data

The weather radar data used in this study consist of 24 hydrological years (September–August), between 1990–1991 and 2013–2014, observed by the Electrical Mechanical Services (EMS/Shacham) non-Doppler C-band weather radar (5.35 cm), located at Ben Gurion Airport (Fig. 1; 31.998°N, 34.908°E). Its effective range is 185 km. Raw radar reflectivity data were translated to quantitative precipitation estimates (QPEs) by initially using a fixed Z–R relationship ($Z = 316 \cdot R^{1.5}$) and then applying physically based corrections and gauge-based adjustment procedures (see details in Marra and Morin, 2015). These produced QPEs at 1-$km^2$, $\sim 5$-min resolutions. Examining the radar QPE and comparing it with rain gauges at hourly and yearly resolution yielded a root mean square error of 1.4–3.2 $mm\,h^{-1}$ and 13–220

$mm\,year^{-1}$, respectively, and a bias of 0.8–1.1 (hourly) and 0.9–1.1 (yearly) (Marra and Morin, 2015). This archive has been previously used for a series of studies focusing on high-intensity precipitation, including precipitation frequency analysis (Marra et al., 2017; Marra and Morin, 2015), floods (Rinat et al., 2018; Zoccatelli et al., 2019), and characterisation of convective rain cells (Belachsen et al., 2017; Peleg et al., 2018). A few issues potentially affecting the QPE should be mentioned. The radar was turned off during the dry season and, for technical reasons, sometimes during the wet season; thus, a few severe storms were missed and are not included in the archive. A long-term decline in the availability and quality of radar data might have decreased the number of high-quality archived HPEs over the years, mainly since 2010. Since we did not aim to provide a complete climatology, these aspects were not expected to influence the results of the study. Due to technical reasons, the radar products were not always available at their intended temporal resolution ($\sim 5$ min) and longer gaps may exist between consecutive radar scans. Gaps of <20 min between consecutive radar scans were linearly interpolated to recreate the 5-min resolution; gaps of >20 min were treated as missing data. Due to the uneven spatial distribution of the rain gauges, adjustment procedures may inadequately represent the southeasternmost areas covered by the radar, where the gauge network is most sparse. Finally, due to overshooting of the radar beam, precipitation occurring east of the Dead Sea (Fig. 1) is generally underestimated.

### 3.2 WRF model configuration

The WRF model was configured using three 1-way nested domains, with a 1:5 resolution ratio between them (Fig. 1) and 68 vertical levels (model top is at 25 hPa). The inner domain ($551 \times 551$ pixels) was set at a 1 $km^2$ horizontal resolution, to be comparable with the radar data. To comply with the Courant–Friedrichs–Lewy numerical stability criterion, model time steps in the innermost domain were 4–8 s (Warner, 2011). However, to spare computer storage, outputs were saved at 10-min intervals. When analysed, the WRF grid was interpolated using nearest-neighbour interpolation from a Lambert projection grid to a similar-sized grid on a transverse Mercator projection, as in the radar archive. It is important to note that a 1 $km^2$ spatial resolution enables explicitly resolving convection, without the use of parametrisation (e.g., Prein et al., 2015). The two outer domains used the WRF Tiedtke scheme for the parametrisation of convection (Tiedtke, 1989; Zhang et al., 2011a). The model input data were 6-hourly ERA-Interim reanalyses, at $\sim 80$ km horizontal resolution and with 60 vertical levels, including sea surface temperature, along with basic meteorological parameters (Dee et al., 2011). The model was used to simulate the HPEs identified in the radar archive (Sect 3.3; Table S1). Each simulation started 24 h prior to the beginning of the event, rounded down to the previous 6 h, and stopped with

the ending of the HPE, rounded up to the next 6 h. Therefore, the spin-up period of each simulation was at least 24 h. Additional model settings, presented in Table 1, were selected because they are considered suitable for convection-permitting simulations (e.g., Romine et al., 2013; Schwartz et al., 2015).

### 3.3 HPE identification

HPEs have various definitions in different research fields and geographical regions. For example, climatologically, HPEs are commonly associated with a specific time interval (i.e., sub-daily to a number of consecutive days) during which precipitation depth surpasses a threshold representing a predefined quantile (e.g., $95^{th}$ or $99^{th}$), or a high, but constant intensity (e.g., 10, 20, or 50 $\mathrm{mm\,day^{-1}}$) (e.g., Drobinski et al., 2014; Nuissier et al., 2011; Westra et al., 2014; Zhang et al., 2011b). On the other hand, hydrological definitions usually focus on the resulting flood. In general, a good definition of a HPE should also include the areal dimension, to enable considering hydrological and social impacts (Easterling et al., 2000).

Here we define HPEs by the exceedance of local, quantile-based thresholds over a sufficiently large area. The decision to set local thresholds was due to the sharp climatic gradient characterising the study area. To decrease the computational effort and guarantee adequate temporal sampling, the HPE identification was based on a radar database comprising hourly intervals for which at least 60% of the expected radar scans are available (Marra et al., 2017). For a set of durations between 1 and 72 h, we defined the threshold as the $99.5^{th}$ quantile of the non-zero (i.e. >0.1 mm) hourly amounts observed in each radar pixel. The range of examined durations was chosen to represent both short and long span HPEs. It should be noted that the same storm can be identified as a HPE for multiple durations. Depending on duration and location, the obtained amounts are equivalent to annual return periods of roughly 2–10 years (Fig. 2). To account for the spatial scale, we classified all time intervals during which at least 1000 pixels (i.e., 1000 $\mathrm{km^2}$) exceed their local threshold as HPEs. Jointly, these thresholds (99.5% for each pixel, and aggregation of 1000 pixels for an event) settle the trade-off between having too many (or too few) events and accounting for HPEs that are too local (or only including the most widespread rainstorms). These selected thresholds enable analysing a reasonable number of diverse HPEs, with some being quite local and others more widespread.

The selection procedure yielded 76–98 individual events for each of the examined durations, summing to 120 when overlaps between durations were included. Similar to Marra and Morin (2015), storms were separated by at least 24 h with <100 pixels displaying rainfall of >0.1 mm. Since the ERA-Interim data are available at 6-h resolution, rainstorms that were too short (<12 h) were excluded from the analysis. Storms longer than 144 h were excluded to avoid

major changes in sea surface temperature during events. In addition, events were discarded manually when the radar data were abundantly contaminated by ground clutter due to anomalous propagation, or when other data-quality issues were observed. The final list of HPEs consisted of 41 independent events spanning on average $3.4 \pm 1.6$ days (Table S1).

For each of these events, a filter was used to remove pixels with residual ground clutter. Pixels in which the probability of rain detection (POD, i.e., the fraction of time in which the pixel exceeds 0.1 $\mathrm{mm\,h^{-1}}$) exceeds 10% and is larger than 1.9 times the average POD of the surrounding area ($25 \times 25$ km) were removed. The extent of the explored area and of the ratio were chosen subjectively after examining ranges between 1 and 3 (for the ratio) and 5 and 50 km (for the areal extent). Additional areas known to be persistently contaminated by ground echoes (from our experience and earlier studies) were masked out manually (e.g., the circular area near the radar). Together, these procedures excluded $\sim 0.5\%$ of the radar pixels.

### 3.4 Synoptic classification

We classified the HPEs into two classes representing the most common rainy synoptic circulation patterns prevailing in the region: MC and ARST. To do so, we relied on the semi-objective synoptic classification by Alpert et al. (2004), based on daily (at 12 UTC) meteorological fields at the 1000 hPa pressure level from the NCEP/NCAR reanalysis ($2.5°$ spatial resolution). We classified a HPE as a MC if one of the following conditions occurred: (i) most of the days comprising the HPE were considered, according to Alpert et al. (2004), as days with either a MC or a high-pressure system following a MC; (ii) one of the days during the HPE was a MC and none of them was an ARST. Similarly, we classified a HPE as an ARST if (i) most of its days were classified as ARST according to Alpert et al. (2004), or (ii) one of its days was an ARST and none of them was a MC. The abovementioned Tropical Plume synoptic pattern (Rubin et al., 2007; Tubi et al., 2017) is not part of our classification because of its low frequency and because it does not appear in near sea level pressure meteorological fields. Specifically, one HPE (#41; Table S1) was characterised, during its 5-day span, first by the prevalence of a Tropical Plume (Armon et al., 2018) and then by a MC; it was classified here as a MC. Despite the simplification, these two classes have been recently shown to exhibit distinct characteristics of rainfall intensity distribution (Marra et al., 2019). Indeed, 85% and 15% of HPEs were classified as MCs and ARSTs, respectively (Table S1), reasonably following the expected proportions of the two synoptic circulation patterns (Goldreich et al., 2004; Saaroni et al., 2010).

## 3.5 Evaluation of simulated rain fields

Inaccurate initial conditions in the presence of non-linear precipitation-generation processes, together with the presence of atmospheric instabilities, may limit the atmospheric predictability and, consequently, modelling skills (Anthes et al., 1985). Moreover, increasing the model resolution may pose difficulties in a pixel-by-pixel evaluation of the forecasts (e.g., Davis et al., 2006; Mass et al., 2002). Approaches that are more suitable for high-resolution rainfall fields range from simple visual comparisons to more sophisticated, object-oriented or filtering methods capable of representing spatiotemporal properties of the fields (e.g., Davis et al., 2006; Gilleland et al., 2009; Roberts and Lean, 2008). In this study, we applied visual comparisons and several numerical measures to compare the observed radar QPE with the WRF-derived rain field.

### 3.5.1 Fractions Skill Score

To evaluate rainfall accumulation for different neighbourhood sizes (namely, spatial scales), we used the method suggested by Roberts and Lean (2008). The methodology includes a conversion of the continuous rain field to a binary field based on the exceedance of a given rain-depth threshold. The fraction of model-output positive pixels (i.e., pixels that have exceeded the threshold) within a certain neighbourhood size is then compared with the matching fraction from the radar QPE, through the Fractions Skill Score (FSS) statistic (Supplementary material [S1]). When the forecast is perfect and unbiased, i.e., when an equal number of observed (in our case, radar) and forecasted (WRF) pixels exceed the threshold, $FSS = 1$. If there is a bias, the FSS will tend asymptotically to a lower value. To quantitatively evaluate the model's ability to predict the observed rainfall above the selected threshold, within a close-enough distance, the uniform FSS (halfway between a random forecast and a perfect skill forecast, yielding a hit rate of 0.5; [S1]) is also calculated. An FSS that is larger than the uniform FSS is considered skilful. It is important to note that if the FSS exceeds the uniform FSS on too large a spatial scale, the forecast might still be skilful, but it is not useful. We applied the FSS method to the cumulative rain field, comparing the radar QPEs and WRF rainfall output (Sect. 4.3).

### 3.5.2 Structure–amplitude–location analysis

To evaluate the characteristics of the WRF precipitation forecast errors, we used the object-oriented structure – amplitude – location (SAL) analysis (Wernli et al., 2008) (Supplementary material [S2]). As in the FSS analysis, it was applied to the cumulative rain field. The SAL analysis splits the rain field into three distinct components and yields a skill score for the forecast of each of them; in each of the components, a zero score indicates a perfect forecast. The amplitude component (A) expresses the model's over/underestimation of the total rainfall for a specific rainstorm (with $A \in [-2, 2]$, and $A = 1$ or $A = -1$ indicating over and underestimation by a factor of 3, respectively). The location component ($L \in [0, 2]$) sums the differences between modelled and observed (i) centre of mass of precipitation and (ii) average distance between the centre of mass and the location of precipitation objects that constitute the rain field (i.e., connected regions in which the cumulative rainfall exceeds 1/15 of the maximal cumulated value; Wernli et al., 2008). The structure component ($S \in [-2, 2]$) quantifies the tendency of the forecasted precipitation objects to be either too smooth (positive values) or too noisy (negative values) relative to the observations.

### 3.5.3 Depth–area–duration curves

Areal rainfall amounts are crucial drivers of the hydrological response and are important for understanding rainfall structure and triggering mechanisms (e.g., Armon et al., 2018; Durrans et al., 2002; Kalma and Franks, 2003; Zepeda-Arce et al., 2000). To quantify and compare observed and simulated areal rainfall amounts, we used depth–area–duration (DAD) curves, which represent the areal extent for which given rainfall depths over given durations are exceeded (Zepeda-Arce et al., 2000).

### 3.5.4 Autocorrelation structure of rain fields

High-intensity, small-scale convective rain cells are among the main factors generating flash floods in small, mountainous and desert catchments (e.g., Armon et al., 2018; Doswell et al., 1996; Merz and Blöschl, 2003), and their fine spatiotemporal structure directly affects the potential of raingauge monitoring (Marra and Morin, 2018). To analyse the convective rain structure, we computed, from both the observed radar QPE and the WRF output, the spatial autocorrelation structure of the maps containing convective elements using the methodology presented by Marra and Morin (2018) (an example is given in supplementary Fig. 1 [SF1]). We interpolated the radar QPEs to 10-min intervals to match the model's temporal resolution, and defined as convective rainfall fields all rain maps in which at least one convective rain cell, defined as a connected region $\geq 3$ km$^2$ with rain intensity exceeding 10 mm h$^{-1}$ and including at least one pixel exceeding 25 mm h$^{-1}$, is observed (Marra and Morin, 2018). We computed the 2-D spatial autocorrelation function of the convective fields following the method in Nerini et al. (2017). A three-parameter exponential function (Eq. 1) was fitted to the 2-D spatial autocorrelation to quantify the correlation distance:

$$r(h) = ae^{-(\frac{h}{b})^c} \tag{1}$$

where $h$ is the lag distance, $b$ is the correlation distance (the distance at which the correlation drops to $r = e^{-1}$), and $a$ and $c$ are the nugget and shape parameters of the curve, re-

spectively. Equation 1 results in an approximation of the 1-D autocorrelation function of convective rain fields. Spatial heterogeneity of the autocorrelation field is quantified by calculating the deviation of the 2-D autocorrelation field from isotropy, following the approach in Marra and Morin (2018). To that end, we defined the ellipticity of the 2-D autocorrelation as the ratio of the minor to major axis of the (approximated) ellipse encompassing the $r = e^{-1}$ region of the spatial autocorrelation field (Fig. SF1).

The temporal autocorrelation is computed by converting the 2-D spatial domain to a 1-D array and adopting time as the second dimension, as proposed by Marra and Morin (2018). It is worth noting that the computed temporal correlation distance neglects advection (Eulerian perspective), and is therefore shorter than the correlation distance obtained in a Lagrangian perspective.

## 4 Results

### 4.1 Quasi-climatology of HPEs

Of the 41 identified HPEs, 35 occurred during MC synoptic prevalence and the rest during ARST prevalence. Despite the dependence of the identification on the quality and availability of the radar data, our analysis can be considered "quasi-climatological", as the selected HPEs do not exhibit obvious biases with respect to the rain climatology in the region: (i) their seasonality follows the seasonal pattern of EM rainy days (Fig. 3), although HPEs occur more frequently at the beginning of the winter, presumably due to the high sea surface temperatures; (ii) HPEs are identified throughout the radar archive (with zero to seven HPEs per year); (iii) the frequency of the prevailing synoptic circulation patterns during HPEs (Table S1) resembles the frequency observed on rainy days (Marra et al., 2019); and (iv) HPEs characterised by ARST prevalence are common only during the transition seasons (Fig. 3) (e.g., De Vries et al., 2013).

For most examined durations, rain amounts defining the HPEs are larger near the Mediterranean coast, extending a few kilometres off- and on-shore (Fig. 2). This resembles the observed pattern of high rain intensities near the coast, rather than inland (Karklinsky and Morin, 2006; Peleg and Morin, 2012; Sharon and Kutiel, 1986), also reported for extreme precipitation quantiles observed from both weather radar and satellite sensors (Marra et al., 2017). In contrast, short durations (<12 h) exhibit the highest rain intensities in the arid portions of the region. The frequency of rain in the arid areas is lower than in the rest of the region (Goldreich, 2012); thus, the 99.5% quantiles are based on fewer data. Nevertheless, the reported higher extreme rain amounts for shorter durations are in agreement with previous studies, which showed that highly localised convective rainfall is more common during HPEs in the desert than in other climatic environments in the EM (Marra et al., 2017; Marra and Morin, 2015; Sharon,

1972). In the mountains, the opposite case is seen; rainfall is produced more significantly through stratiform (or shallow convection) processes and therefore, rain amounts for short durations are relatively lower (Sharon and Kutiel, 1986). For the longer durations, rain intensities in the mountains are comparable to the intensities near the coast, probably resulting from the tendency of rain to persist in orography-affected regions (e.g., Panziera et al., 2015; Tarolli et al., 2012).

Affected by higher rain intensities, the centre of mass of the precipitation field for each one of the HPEs is located near the EM coastline (Fig. 4). Nevertheless, a seasonal pattern appears, with a general landward shift of the centre of mass during the rainy season (Fig. 4). This is caused by land–sea differential heating and heat capacities, and resembles the seasonal pattern of rain intensities in the EM (Goldreich, 1994; Sharon and Kutiel, 1986). In fact, this points out the observed preference of convective clouds to form above high-temperature surfaces, i.e., the sea surface or nearby coastal plains in autumn or early winter, and farther inland in the spring. In terms of seasonality, the WRF-simulated centres of mass exhibit a similar, even if slightly less obvious, landward pattern. It must be noted that ARST-type events in the WRF results are biased eastwards, compared to the radar results, which could be related to the WRF's worse performance in such events (e.g., Sect 4.3). Moreover, the exact location of the radar observed centre of mass can suffer from range degradation, which may cause these centres to be biased towards the radar location.

According to the definition applied in this study, a given event can be considered a HPE for more than one duration. This can happen when the thresholds associated with the examined durations (Sect. 3.3) are exceeded either at the same location, or when they are exceeded in different regions. The durations associated with each HPE are listed in Table S1. The co-occurrence of each HPE duration with the rest of the examined durations is shown in Figure 5; these co-occurrence values are similar to values determined in the Alps by Panziera et al. (2018). For example, 79% of the HPEs at 24 h duration are also HPEs at 72 h duration. Figure 5 indicates a high dependence (i.e., co-occurrence) of the short-duration HPEs (3–12 h). Similarly, there is a high dependence within the long duration (24–72 h) HPEs. Nevertheless, even the shortest duration HPEs examined here show a rather high co-occurrence with the longest duration HPEs (probabilities in all cases $\geq 0.5$).

### 4.2 Bias

Figure 6 shows the rainfall accumulated during all HPEs as estimated by the weather radar, modelled by the WRF, and measured by rain gauges (Fig. 6a, b and d, respectively). Bias, defined herein as the normalised difference between WRF rainfall and radar QPE ($\frac{WRF-radar}{radar}$), in percent, is shown in Fig. 6c. In 69% of the studied region, the bias lies between +200% and -67%, while some areas show a strong

positive bias (Fig. 6c). The three stations highlighted in the figure (the values shown for radar and WRF represent the average of the 9 pixels surrounding the gauge locations) show how this large bias is mostly caused by radar underestimation. In fact, these areas are generally located far from the radar or in the eastern portion of the radar coverage, where radar QPE suffers from range degradation and beam overshoot due to the presence of mountains. In some other areas, the bias seems related to residual beam blockages. Underestimation (bias $< 0$) is also apparent in regions with ground clutter, and some spatial inconsistencies related to the interpolation of a few fully blocked beams can also be noticed. To avoid interference of these radar estimation inaccuracies with our results, we focus only on the areas in which the bias lies between +200% and -67% (Fig. 6c). Still, a portion of the area close to the radar is characterised by negative bias, which could be attributed to simulated rain intensities which are too low. A similar pattern was also shown in Rostkier-Edelstein et al. (2014) where it was attributed to intensities that are too low during deep MCs.

### 4.3 Visual, neighbourhood and object-based evaluation of WRF model simulations

Visual comparison of observed (radar) and simulated (WRF) rainfall fields yielded mostly (subjectively) good results in terms of the spatial rainfall patterns, such as widespread vs. localised rainfall. Figure 7 presents, as an example, a well-simulated HPE case (event #1, Table S1). In addition, the distributions of rainfall among pixels were generally well represented (Fig. 7d). At the same time, pixel-based comparisons were deemed inappropriate for such an analysis, as shown in the scatter plot (Fig. 7e). Most of the examined HPEs led to similar observations, with the exception of two HPEs in which the WRF model clearly failed to represent the rainfall patterns. An example of such a poor simulation is given in Fig. 8 (event #5, Table S1). Both of these poorly simulated HPEs were characterised by relatively short total storm spans (1.7 and 2 days), just exceeding the durations that defined them as HPEs (6 h and 3–24 h, respectively). Synoptically, they were classified as ARSTs, a system generally characterised by local, short-lived convection associated with a localised rainfall-triggering mechanism (Armon et al., 2018). The skill of mesoscale models (e.g., WRF) is poorer in simulating these types of events, mainly due to their short predictability and stochastic nature (see e.g., Yano et al., 2018). Although a deeper understanding of these aspects can be beneficial for improving future simulations, it falls outside the scope of this study and requires future dedicated research efforts.

The FSS of the first HPE (Fig. 7f) further manifests the accuracy of the simulated rainfall fields. The forecast has a larger FSS than the uniform FSS for all of the examined cumulative rainfall amounts $\leq 50$ mm, even at the model resolution (1 km). For larger cumulative rainfall the FSS is unstable due to a limited amount of observations of such cases, and thus, no conclusions about the goodness of the results can be made for the occurrence of such high cumulative rainfall. It is only for the higher rainfall amounts, e.g., 125 mm, corresponding to less than 1% of the pixels in this HPE, that the model forecast is unskilled at all spatial scales, i.e., the uniform FSS outperforms the WRF forecast FSS (yet, these results are based on a limited amount of data as well).

During EM rainstorms, cumulative rainfall values are distributed unevenly in space, and extremely high rainfall depths are embedded within the larger aerial coverage of lower rainfall depths (e.g., Armon et al., 2018; Dayan and Morin, 2006; Morin et al., 2007). Forecasting the spatial distribution (location and spatial frequency) of low cumulative rainfall is thus easier than forecasting the distribution of the high end of cumulative rainfall, even when averaging is conducted over large scales. The minimal scale (Roberts and Lean, 2008) at which the FSS of the model's forecast is higher than the uniform FSS was calculated for the occurrence of a range (1-200 mm) of cumulative rainfall depths, for all of the identified HPEs (Fig. 9). This allows estimating the minimal scales for skilful rainfall detection for rain depths that equal or are greater than an arbitrary cumulative rain depth threshold. For example, the original model resolution yielded a skilful forecast for the occurrence of cumulative rainfall depths of $< 25$ mm in 50% of the HPEs (Fig. 9). The figure also shows that the occurrence of cumulative rainfall exceeding 45 mm, in most cases, is skilfully forecasted only on a relatively large spatial scale (tens of kilometres). During ARSTs, the minimal scale was much higher than during MCs (not shown); however, it is important to remember that two of these HPEs were poorly simulated.

The SAL analysis (Fig. 10) showed good performance of the model, except for a substantial positive amplitude bias (inter-event amplitude component median $= 0.80$ [i.e. bias of 130%, as defined in Sect. 4.2], interquartile range $= 0.37 - 1.02$). Two events stood out with a bias smaller than zero; these were the abovementioned poorly simulated HPEs. In general, MC-type HPEs exhibited much greater bias than ARSTs (inter-event amplitude component median $= 0.85$ versus $-0.07$). However, it must be noted that the median of ARST-type HPEs includes the two poor simulations, and is therefore predicted to be more negative. Surprisingly, where visual comparisons seemed better, and the structure component was closer to zero, the amplitude component actually suffers from more positive biases. E.g., the structure component of event #1 (Fig. 7, Table S1) is 0.04, while its amplitude component $= 1.03$; furthermore, the median amplitude of events characterised by a structure component larger than the median structure is 27% higher than the amplitude of events with a structure smaller than the median value.

The structure component was well modelled in most cases, showing the ability of the WRF to accurately generate precipitation objects (0.06 and -0.06 to +0.26, median and interquartile range, respectively; 0.05 and 0.09 are the medians

for MC- and ARST-type events, respectively). This is particularly important in regions and events where rainfall is generated through both convective and stratiform processes, or when intense rainfall is embedded within larger-scale low-intensity precipitation (Wernli et al., 2009). The slight positive tendency of the structure component could either indicate that the model creates rain fields that are too smooth (lower intensity rain, and objects that are too large), or that radar data are too noisy, or most probably, a combination of both error sources.

Relatively low values of the location component (0.25 and 0.18–0.31, median and interquartile range, respectively; 0.26 and 0.22 are the medians for MC- and ARST-type events, respectively) demonstrate the model's high capability to spatially distribute precipitation objects. Medially, 34% of this component is composed of the error in the centre of mass location (i.e., a median error of 30 km in the location of the centre of mass), and the rest is from the average location of each precipitation object. Namely, the model prediction of the centre of mass of the rain field is quite satisfying, but the prediction of individual precipitation objects is a bit poorer. The contribution from the $L_2$ component to the location component (Supplementary material [S2]) indicates that modelled precipitation objects are not distributed the same as the observed ones. This is probably due to a mismatch in positioning of the simulated cells. Given the good ability of the radar to represent location of rain cells, attenuation and range-degradation of the radar data should have only minor effects on $L_2$. In either case, location values are rather small, exhibiting good spatial distribution of precipitation objects. Nonetheless, standing out with high location values (0.46 and 0.85) are the same two challenging ARST-type HPEs for which the model was unable to simulate the rainfall in a satisfying manner, yielding large spatial inconsistency with respect to observations (see above, e.g., Fig. 8).

The overall positive bias seen in the amplitude component (Fig. 10) could result from underestimation of the radar QPE or overestimation of the WRF simulation. Possible reasons leading to radar underestimation were discussed above, and may contribute to this bias even after the most severely biased regions have been masked. However, this positive bias still needs to be considered when addressing the actual cumulative rainfall amounts predicted by the model. In contrast with the overall bias (Sect. 4.2) almost no event showed a negative bias. Positive biases were attributed by Rostkier-Edelstein et al. (2014) mainly to deep lows over complex-terrain regions.

The overall good representation of precipitation objects implies that precipitation processes generated by the model represent actual processes and rainfall characteristics (Wernli et al., 2009).

## 4.4 Characterisation of rainfall patterns

### 4.4.1 Areal rainfall

Figure 11 shows the depth-area-duration (DAD) curves obtained from all 41 HPEs for durations of 30 min, 6 h and 24 h from radar QPEs (Fig. 11a, c, and e, respectively) and WRF (Fig. 11b, d, and f, respectively). When referring to DAD analysis, the term "duration" represents the time period, over the course of each HPE, where maximum rainfall depths were observed. A major increase in cumulative rainfall with increased duration is observed for both the radar and WRF curves (Fig. 11g): e.g., based on the radar, an area of $10^3$ km$^2$ is medially covered by 9 mm for a duration of 0.5 h, which increases to 35 mm and 60 mm for 6 and 24 h, respectively (corresponding values from the WRF-derived rainfall are 4, 25 and 50 mm). This increase could be explained by either continuous rainfall or frequent arrival of rain cells into the region. The latter increases the wet area and the cumulative rainfall in areas that have already experienced rainfall, and is a major characteristic of HPEs in the EM (e.g., Armon et al., 2018, 2019; Sharon, 1972). Furthermore, over the longer durations, this causes DAD curves for different events to be more similar to one another (e.g., Fig. 11e and f).

The inter-event spread and the difference in the DAD curves for MC and ARST (Fig. 11a–f) illustrate the various types of HPEs identified here. These types range between rainstorms exhibiting only a minimal increase in rainfall area with time, i.e., almost all of the rainfall precipitates during a short period, and rainstorms composed of many rain cells passing through the same area, or long-lasting rainstorms. These results confirm previous findings by Armon et al. (2018) based on a more limited number of events: HPEs classified as ARSTs (Table S1) tend to be of higher rain intensities for smaller regions and shorter periods than HPEs classified as MCs. MCs only exhibit higher rain intensities over larger regions and for longer durations.

It is important to note the difference between radar QPE- and WRF-derived rainfall DAD curves. Higher rain values in the radar QPE over the range of smaller areas is the most obvious difference (Fig. 11g). Although these higher values may, at first glance, indicate that the WRF is unable to reproduce the high-intensity rainfall of the HPEs in the EM, it should be remembered that at short durations, high-intensity radar QPEs can be of lower accuracy due to contamination from residual ground clutter or hail. This may affect the QPEs of the smaller areas more selectively. For instance, for one of the HPEs, an area $> 100$ km$^2$ has a rain amount $\geq 100$ mm in 0.5 h (Fig. 11a), a value that exceeds the 200-year return period for the area (Morin et al., 2009). Other notable differences are some ARST-classified HPEs with WRF-derived DAD curves (Fig. 11b, d, and f) consisting of the two WRF-unresolved HPEs mentioned above, and yielding a median ARST curve that is much lower than the radar-derived one.

The reported differences between WRF- and radar-derived curves result in an overall greater area-over-threshold radar curves for the high-rainfall thresholds, especially for the short durations. For long durations and low rainfall thresholds, the WRF area is larger (Fig. 11), reflecting the positive bias mentioned above.

### 4.4.2 Autocorrelation structure of convective rainfall

HPEs in the EM are commonly composed of highly localised convective rain cells. This is well reflected in the sharp decrease of the 1-D autocorrelation describing the convective rain fields (Fig. 12a and b) obtained using all of the convective rain fields throughout the 41 HPEs ($n = 11,731$ snapshots for radar and $n = 14,323$ for WRF). The median decorrelation distance (defined as the distance in which the correlation drops to $r = e^{-1}$, i.e., parameter b of the 1-D exponential fit [Eq. 1]) of all convective rain snapshots from the radar data is 9 km (7 km using the WRF-derived rainfall) and ranges between 3 and 23 km (for the 10% and 90% quantiles, respectively; 2 and 20 km using WRF). The median decorrelation distance during ARSTs is shorter than during MCs, as obtained from both the radar (7 km and 10 km, respectively for ARSTs and MCs) and the WRF (5 km and 7 km, respectively). These values are comparable to previously reported observations (e.g., Ciach and Krajewski, 2006; Morin et al., 2003; Peleg and Morin, 2012; Villarini et al., 2008) and are somewhat larger than the reported values for the southeastern part of the area by Marra and Morin (2018). However, it should be noted that Marra and Morin (2018) examined 1-min rainfall fields versus the 10-min fields examined here.

The median of the temporal decorrelation distance (Fig. 12c and d) was $\sim 4$ min ($\sim 14$ min for the WRF), and it ranged between $< 1$ and 19 min (10% and 90% quantiles, respectively; 3 and 29 min using WRF). Despite agreeing with the results of Marra and Morin (2018), the exact temporal decorrelation distance is somewhat dubious, since it is shorter than the time step used for its calculation (10 min). For this reason, we do not report the small differences that exist between the two synoptic systems. The larger temporal correlation in the WRF-derived rainfall is expected, because radar QPE suffers from temporal inconsistencies (e.g., when a convective cell passes through a region with beam blockages). Nevertheless, such a short temporal decorrelation confirms the local and spotty nature of rainfall characterising HPEs in the region.

The declining pattern of the 1-D autocorrelation overlooks the 2-D spatial heterogeneity of the autocorrelation field. The ellipticity of the 2-D autocorrelation yielded a median across all convective rain fields value of 0.56 (0.62 and 0.54 in ARST- and MC-type events, respectively), with a range of 0.33–0.80 (10%–90% quantiles). WRF-derived ellipticity values were almost the same: 0.58 (0.68 and 0.68 in ARST- and MC-type events, respectively), with a range of 0.33–0.79. These autocorrelation ellipses in the radar data were oriented 13° anti-clockwise from the east-west axis (median value; 7° and 14° for ARST- and MC-types, respectively) and 22° for the WRF ellipses (10° and 24° for ARST- and MC-types, respectively). These values are similar to the orientation of radar rain cells in the eastern part of the region (Belachsen et al., 2017), but somewhat different from the orientation of autocorrelation fields from the south-eastern part of the region (Marra and Morin, 2018). Orientations found in the present analysis cover the entire evolution of HPEs and thus include both south-west (mainly at the beginning of the storm) and north-west (mainly at the end of the storm) alignments of rain cells. Therefore, they are oriented more anti-clockwise than the autocorrelation fields from the south-eastern part of the region (Marra and Morin, 2018), which commonly represents rainfall at the end of a rainstorm (Armon et al., 2019). Moreover, Marra and Morin (2018) examined 1-min snapshots while here, advection can play a role in the examined 10-min time interval. Finally, Marra and Morin (2018) analysed only 11 events, thus, inter-event variance may still play a large role in their results. The high agreement between modelled and observed rain field ellipticity and orientation also demonstrates the high skill of the WRF simulations in accurately representing convection in the region and thus, reproducing rain-cell properties.

### 4.5 Summary of results

This work characterises rainfall patterns during 41 HPEs in the EM and evaluates the ability of a high-resolution WRF model to properly simulate their cumulative rain field and spatiotemporal behaviour, with a specific emphasis on their convective component and the prevailing synoptic system. A successful outcome will open the way to downscaling global climate projections to induced changes in rainfall patterns on a regional scale during HPEs, with an understanding of the strengths and weaknesses of the regional results. However, it is important to note that identification of HPEs in global climate models constitutes yet another challenge (see discussions e.g., in Chan et al., 2018; Gómez-Navarro et al., 2019; Meredith et al., 2018).

To overcome the diverse climatology of the EM, we identified HPEs using pixel-based weather radar climatology. We used a uniquely long, quality-controlled and gauge-adjusted high-resolution weather radar archive to characterise the rainfall patterns. A convection-permitting high-resolution WRF model configuration was used to simulate the same HPEs and the results of this modelling effort were compared to the radar QPEs. For most of the 41 HPEs, model simulations gave valuable results: using the FSS we determined that (i) WRF simulations are highly accurate for cumulative rainfall $< 25$ mm (Fig. 9; Sect. 4.3), (ii) accumulation of $> 45$ mm produces variable results among different cases (Figs. 7, 8 and 9; Sect. 4.3). In other words, skilful results are gained if the model output is averaged over at least a few tens of kilometres. SAL analysis of cumula-

tive rainfall showed that rainfall location and structure were correctly reproduced by the model and were similar to the weather radar data observations in 39 out of the 41 HPEs. Conversely, rainfall amplitude was highly (positively) biased, with some of the bias likely due to radar underestimation; however, a model positive bias cannot be excluded. Furthermore, we found that ARST-type HPEs are simulated worse than MC-type events, at least in terms of FSS and the spatial structure of rainfall.

In general, rain amounts forming HPEs are higher near the EM coastline with the exception of (i) short (examined) durations, for which the highest rain amounts are observed in the desert regions, and (ii) the longer-duration HPEs, for which mountainous rain amounts are comparable to those on the coast. Identified HPEs occurred during the wet season (October–April), primarily in November–February. Their centre of mass was close to the Mediterranean coastline and shifted landward during the season. We analysed the areal distribution of rainfall at various durations, the autocorrelation structure of the convective rainfall fields and DAD curves, to obtain quantitative information on the characteristics of the rainfall fields, the ability of the WRF model to simulate them, and the processes generating them, such as the aggregation of small and short-lived rain cells to produce a HPE.

## 5 Discussion

### 5.1 Spatial distribution of rain-intensity thresholds defining HPEs

High-intensity threshold-forming HPEs near the Mediterranean Sea (Fig. 2) are expected, because of its warm surface waters and high moisture fluxes; they are also apparent in other regions of the Mediterranean (e.g., Dayan et al., 2015; Ivatek-Šahdan et al., 2018; Khodayar et al., 2018; Pastor et al., 2002; Peleg et al., 2018; Tarolli et al., 2012). High rain intensities in the desert are somewhat more intriguing. For example, Warner (2004) mentioned that there are contrasting evidence of whether rain intensities in the desert being higher than in non-desert regions. An opposing trend between mean annual rainfall and short-duration rain intensities was also described by Sharon and Kutiel (1986) using rain gauges, and by Marra and Morin (2015) using both rain gauges and weather radar. This trend is related to the higher surface temperatures in desert regions, which may enhance convective activity (e.g., Peleg et al., 2018), as well as to a deeper boundary layer (e.g., Gamo, 1996; Marsham et al., 2013) and the prevalence of rainfall from ARST circulation patterns, which generally cause higher rain intensities (Armon et al., 2018; Nicholson, 2011; Sharon and Kutiel, 1986; De Vries et al., 2013). Such a sharp spatial change in the climatology of the rain intensities defining HPEs can only be captured using high-resolution, high-spatiotemporal-coverage data (such as the radar QPE presented here), and

reproduced by high-resolution, convection-permitting models.

### 5.2 Multiple-duration HPEs and their relation to flash floods

Mediterranean-climate, and even more so desert-climate HPEs, can produce rain amounts of the same order of magnitude as the mean annual rainfall (e.g., Nicholson, 2011; Schick, 1988; Tarolli et al., 2012). Frequent co-occurrence of short- and long-duration HPEs is thus to be expected, and dividing events into short versus long duration is not straightforward. However, our dataset comprises events with different characteristics: local and intense, as well as widespread; rainfall-triggering mechanisms and potential hydrological impact can be quite different.

Comparison of the DAD curves in Fig. 11 with reported floods in Mediterranean and desert environments in the EM (Zoccatelli et al., 2019) shows that a portion of the HPEs analysed here are prone to produce floods in smaller catchments and in desert regions, characterised by rather short ($\sim$ 7 h) and low total precipitation rain spells. Other HPEs analysed here could generate floods in larger catchments and in Mediterranean climate regions, characterised by longer rain spells and higher rain depths (1 day, 52 mm, respectively) (Zoccatelli et al., 2019). Specifically, the convective part of the rainstorm is known to generate the highest-magnitude floods, even in Mediterranean climate areas (e.g., Rinat et al., 2018; Tarolli et al., 2012). The short spatiotemporal autocorrelation distances observed for the convective rain fields highlight, once again, the spottiness of HPE rainfall in the EM region (Sharon, 1972), and were well-simulated by the WRF model (Fig. 12).

### 5.3 Identification and characterisation of HPEs using weather radar and high-resolution weather model

ARST synoptic circulation is often associated with flash floods in the desert part of the region (Ashbel, 1938; Kahana et al., 2002; Krichak et al., 1997), and its rainfall is commonly caused by mesoscale triggering of convection (Armon et al., 2018) and is therefore less predictable (e.g., Keil et al., 2014), as evident from this study as well (e.g., Fig. 10-11). ARSTs are also characterised by smaller rain field autocorrelation distance (Fig. 12). It is thus crucial for future studies to understand the reasons for the poor modelling results observed with 2 of these 41 HPEs. This is evident in the coarser model domains as well (SF2). Possible aspects to be inspected include the adopted parametrisation schemes (Table 1), but since we used convection-permitting resolution, problems could arise from other issues. In particular, since errors in the moisture field tend to propagate rapidly, the correct amount of moisture must be entered into the model in the correct location to properly reproduce rainfall on the mesoscale (e.g., Rostkier-Edelstein et al., 2014; Zhang et al., 2007). In

this study, we used ERA-Interim reanalysis data ($\sim 80$ km horizontal resolution), which may not be accurate enough to resolve some conditions, but is on the same scale as outputs of global climate models. Future studies should consider using higher-resolution input data, such as the newly released ERA5 data (Hersbach, 2016).

Nonetheless, the autocorrelation structure of the rain fields was in general, for most HPEs, well simulated (Sect. 4.4.2). This suggests that even if an event is less predictable, some of the rainfall characteristics can still be simulated. This result is encouraging in terms of the use of convection-permitting models, e.g., in nowcasting, because it means that wind patterns (determining orientation and ellipticity) are well forecasted.

The use of a long record of radar QPEs enabled us to provide a high-resolution quasi-climatological characterisation of the rainfall patterns during HPEs with a resolution and spatial coverage that cannot be achieved using rain gauges. However, rainfall characteristics could not be adequately retrieved in regions suffering from radar data-acquisition problems. Nevertheless, the resultant skill of the WRF rainfall fields supports its use for representing HPEs in regions that are not well covered by radars. Since the analyses were performed in a region exhibiting a strong climatic gradient, we suggest that similar results be obtained in other parts of the world, at least in areas characterised by similar climates.

The main added value of convection-permitting models is seen in area averages, rather than over small-scale regions (Roberts, 2008). Therefore, over large catchments (e.g., larger than a few hundred square kilometres, as suggested by the minimal scale presented in Fig. 9), their forecasts are expected to be relatively useful and accurate. Nonetheless, the use of a deterministic convection-permitting model is still unsatisfactory for pinpointing the highest observed rain accumulations. Although such models are becoming more common in weather and climate forecasting and research (Prein et al., 2015), they are still not adequate for short-term hydrological applications, such as flash flood predictions. The structure of the high cumulative rainfall is predicted quite well. However, it still suffers from a positive bias, and is not exactly well located (e.g., Figs. 9 and 10). In order to provide better flood predictions, especially for small catchments and for flash flood generation controlled by infiltration-excess, there is a need for more structured approaches, such as ensemble forecasts and data assimilation of meteorological observations (e.g., Diomede et al., 2014; Gustafsson et al., 2018; Hamill et al., 2008; Rostkier-Edelstein et al., 2014). These would provide probabilistic (rather than deterministic) information, and could therefore account for the uncertainty characterising the location in high-resolution models (e.g., Alfieri et al., 2012; Vincendon et al., 2011).

Characterisation of rainfall patterns during HPEs has special significance in the EM: on the one hand, the region suffers from a severe water shortage; on the other, it is prone to devastating floods. Both are predicted to worsen in response to climate change (e.g., Alpert et al., 2002; Kelley et al., 2015; Sowers et al., 2010). Modelling could help understand the effects of climate change on these two aspects but, before assessing the projections for a change in rainfall patterns induced by climate change, we need to consider what aspects of these patterns are still not well-captured by weather models, posing a challenge for future predictions. For example, we showed here that rainfall during ARSTs is less adequately forecasted. These ARST HPEs are known to cause flash floods and, as ARSTs might be occurring more frequently due to global warming (Hochman et al., 2018a), this low predictability should be addressed.

The work presented herein is a step towards better understanding rainfall patterns during HPEs in the EM, and we are currently extending the research to relate specific rainfall patterns to atmospheric conditions at high-resolution, and to analyse how the predicted climate change will affect the rainfall characteristics outlined in this paper. Another research direction worth following would involve combining our procedures with satellite-based climatology. However, to date, satellite products present temporal ($\geq 0.5$ h, mostly $\geq 3$ h) and spatial ($\geq 0.04°$, mostly $\geq 0.25°$) resolutions (e.g., Ashouri et al., 2015; Gehne et al., 2016) that are insufficient to adequately sample the fine-scale properties of convective rainfall fields, particularly in arid areas.

## 6 Conclusions

This study presents the identification of HPEs using a weather radar. These HPEs were then simulated using a high-resolution NWP model and evaluated, focusing on the spatiotemporal patterns of the rainfall fields. The main conclusions of this characterisation and evaluation are:

- HPEs in the EM are common between October and April, and their occurrences are focused in November–February. The HPEs' centre of mass is located near the Mediterranean coastline and moves landward during the rainy season.

- For most examined storm durations, the rain amounts forming HPEs (i.e., larger than 99.5% of all rainy hours) are higher near the Mediterranean coast. For short durations, the highest HPE rain amounts are located in the desert, and for long durations, mountainous and coastal regions exhibit similar values.

- HPEs consist of small convective rain cells (spatial and temporal decorrelation of $\sim 9$ km and $\sim 4$ min, respectively) that form a highly variable rainy area over short durations. The size of the rainy region increases with duration and becomes more homogeneous between events.

- A convection-permitting high-resolution WRF model can simulate most HPEs, apart from some of the short-

est, most localised storms, associated mainly with AR-STs.

– Rainfall structure is well simulated. Nevertheless, it is slightly less variable than the observed structure, and is characterised by a significant positive bias in rain volume. This can be, at least partially, attributed to radar underestimations.

– The location of rainfall is generally predicted properly. About a third of the location error comes from a spatial shift of 30 km in the centre of mass, and the rest from the difference in the location of specific precipitation objects.

– The minimal scale for forecasting total rainfall depths $< 25$ mm is 1 km. It raises to a few tens of kilometres in cumulative rainfall $\geq 25$ mm, and even more for rain depths $> 45$ mm. For such large cumulative rain depths the minimal scale becomes highly variable between events.

Use of a high-resolution weather model that can reproduce rainfall patterns during HPEs is of great importance in predicting the hydrometeorology of flood-producing rainstorms. However, these must be elaborated using, e.g., ensemble runs of the model. Convection-permitting models may also help assess changes in precipitation induced by climate change, although if they are composed of HPEs that are less skilfully predicted in present, they should be examined with caution.

*Data availability.* Rain gauge data were provided and pre-processed by the Israel Meteorological Service (https://ims.data.gov.il/; freely available in Hebrew only). Shacham radar data were provided by EMS-Mekorot projects (www.emsmekorotprojects.com). Corrected and gauge-adjusted data (Marra and Morin, 2015) are available in the form of images, through personal communication with the head of the Hydrometeorology lab at the Hebrew University of Jerusalem, Prof. Efrat Morin (efrat.morin@mail.huji.ac.il). ERA-Interim data were downloaded from the Research Data Archive at the National Center for Atmospheric Research, Computational and Information Systems Laboratory: European Centre for Medium-Range Weather Forecasts, 2012, updated monthly: ERA-Interim Project, Single Parameter 6-Hourly Surface Analysis and Surface Forecast Time Series (https://doi.org/10.5065/D64747WN). The WRF namelist.input file can be found in the supplementary data.

*Author contributions.* MA and EM conceptualised this work. Data curation and formal analysis were performed by MA and FM. Funding was acquired by EM, YE, FM and DRE. EM and YE supervised the work. MA wrote the original draft of this paper, which was reviewed and edited by all authors.

*Competing interests.* The authors declare that they have no conflict of interest.

*Acknowledgements.* This study is a contribution to the PALEX project "Paleohydrology and Extreme Floods from the Dead Sea ICDP Core", funded by the DFG (grant no. BR2208/13-1/-2). It is also partially funded by the Israel Science Foundation (1069/18), the NSF–BSF (BSF 2016953), the Israel Ministry of Science and Technology (grant no. 61792), the Advanced School for Environmental Studies at the Hebrew University of Jerusalem, and the Israel Water Authority, and is a contribution to the HyMeX program. The authors thank Prof. Pinhas Alpert for the updated synoptic classification data. The authors also thank three anonymous reviewers and the editor, Prof. Marie-Claire ten Veldhuis, for their constructive comments, which helped to improve this manuscript.

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

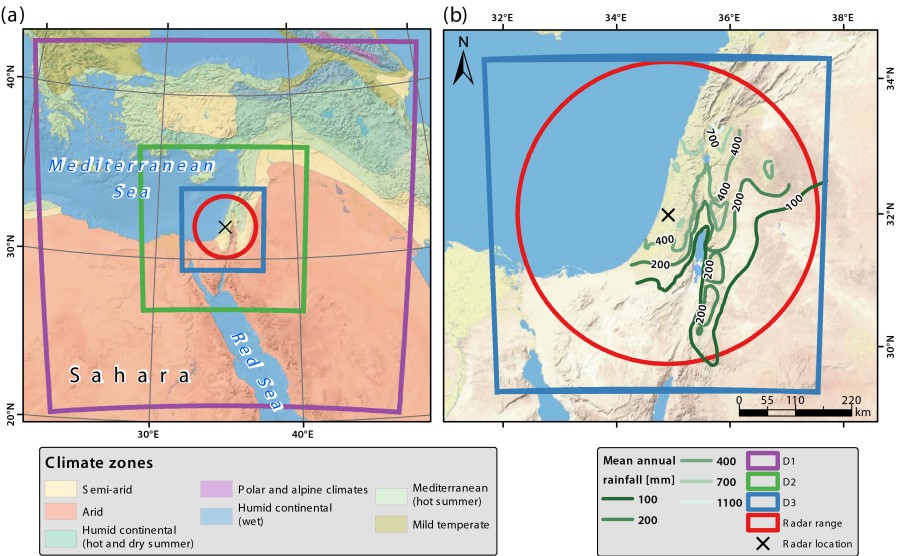

**Figure 1.** Study region. (a) Climate zones in the eastern Mediterranean, three nested domains used in the weather model (D1-3; purple, green and blue) and the radar domain (red). (b) Mean annual rainfall isohyets, radar and innermost model domains. Climatic classification is from the Atlas of Israel (2011). Basemap source: U.S. National Park Service.

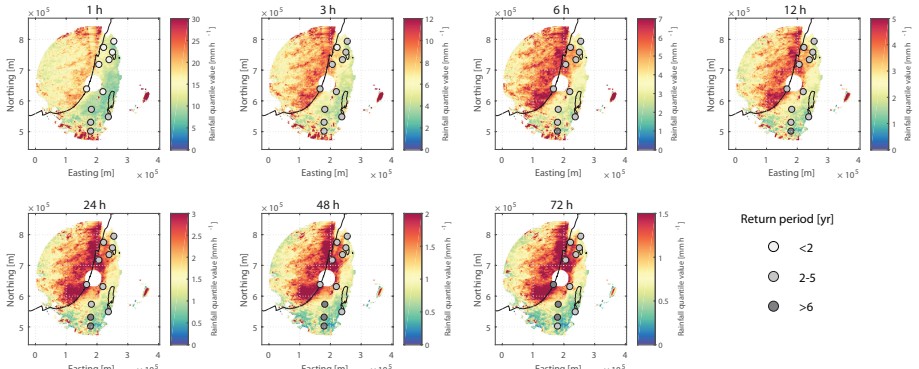

**Figure 2.** The 99.5% rain intensity quantile of each radar pixel for durations of 1 h (top-left) to 72 h (bottom-right). Notice change in colour scale between different durations. Also shown are annual return periods of the rain-intensity threshold averaged over 9 pixels around 11 locations (generalised extreme value fit of the rain gauge annual maxima series, using the method of probability-weighted moments, with records of at least 44 years). These computed annual return periods range between 1.8 and 10.4 years. White areas found mostly to the east of the radar were masked out according to the black line in Fig. 6c (Sect. 4.2).

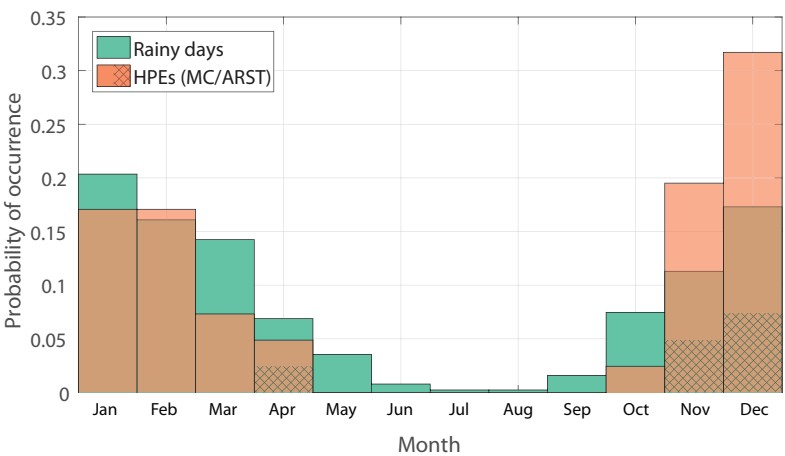

**Figure 3.** Monthly probability of occurrence of rainy days near the radar location (green; Bet Dagan rain gauge, 32.0°N, 34.8°E), and of HPEs from the radar archive (orange). Hatching represents HPEs classified as ARST.

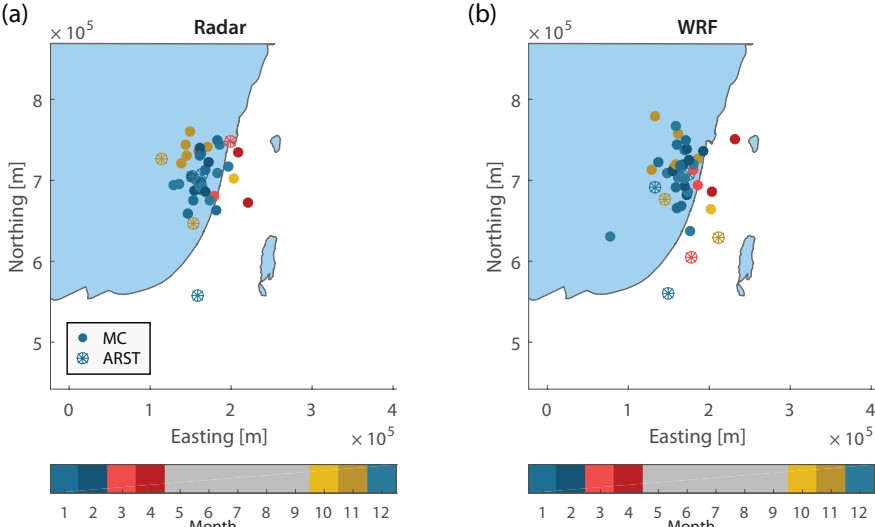

**Figure 4.** Centres of mass of cumulative rainfall of each of the HPEs derived from (a) radar QPE and (b) WRF. Colours represent month of occurrence. Synoptic classification according to Sect. 3.4.

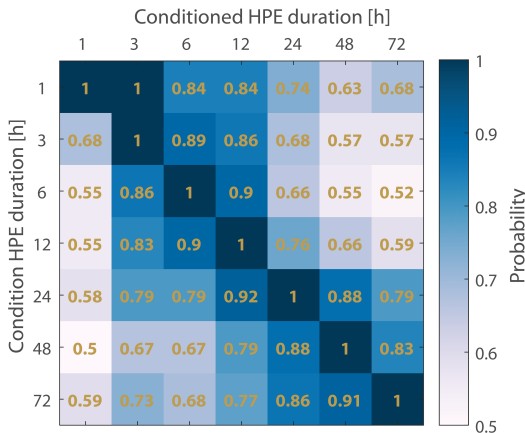

**Figure 5.** Probability of a HPE with a given examined duration listed on the x-axis conditioned on being a HPE with a duration listed on the y-axis.

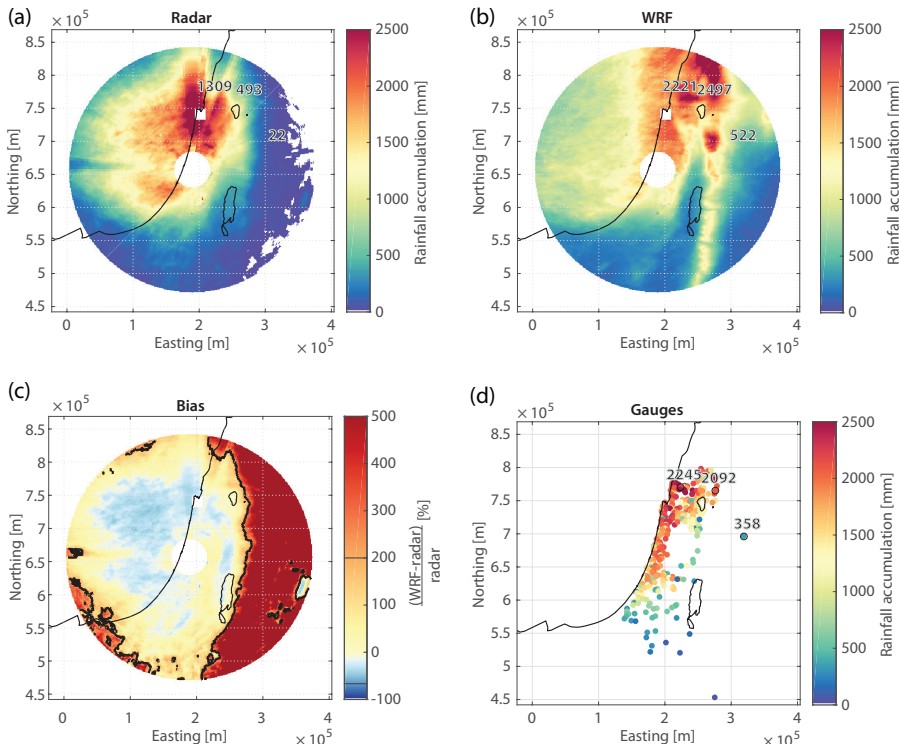

**Figure 6.** Total cumulative rainfall for all 41 HPEs, from (a) radar-derived QPE, (b) WRF-derived rainfall, and (d) daily rain gauges. (c) WRF-to-radar rainfall accumulation bias (normalised difference; Sect. 4.2). The 200% and -67% bias region is marked in black. Highlighted in (d) are total accumulations [mm] measured at three rain gauges from regions where radar QPE is considered to be inferior; corresponding radar and WRF, 9-pixel averaged values [mm] centred over the same locations, are shown in (a) and (b), respectively.

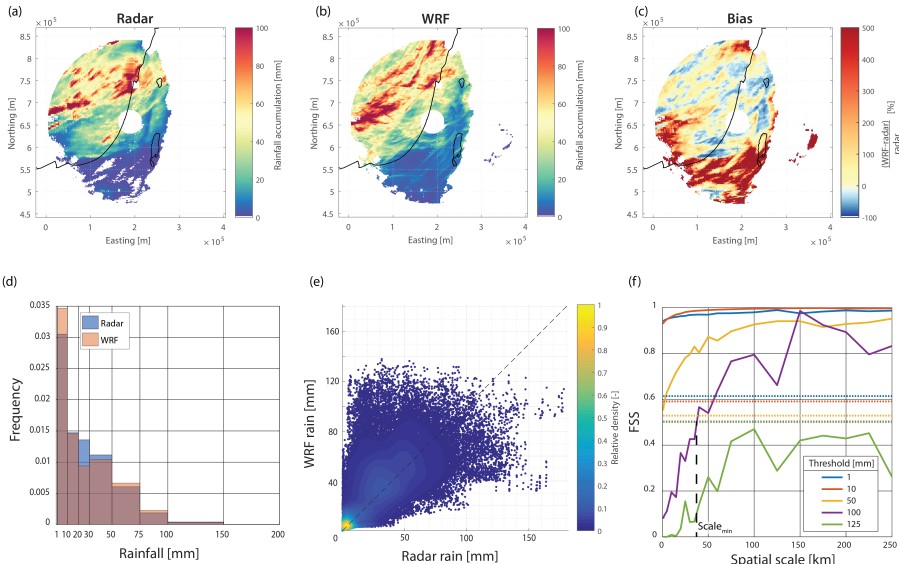

**Figure 7.** HPE #1 (02-Nov-1991 09:00 to 05-Nov-1991 09:00 [local winter time]; see Table S1). Cumulative rainfall from (a) radar-derived QPE, (b) WRF-derived rainfall, and their ratio (c). A pixel-based comparison between rainfall accumulations using a histogram (d; zero rainfall is omitted) and scatter plot (e). Notice that although rainfall distribution is quite well represented (d), results of a single pixel might deviate substantially from the 1:1 line (e; dashed). The fractions skill score (FSS) for the same event for various cumulative rainfall thresholds is presented in panel (f). Dashed lines are uniform FSS for the same rainfall thresholds. Also shown (dashed black line) is the minimal scale for a valuable prediction for 100 mm rain depth (at the crossing of the FSS and the uniform FSS; see details in supplementary material [S1]

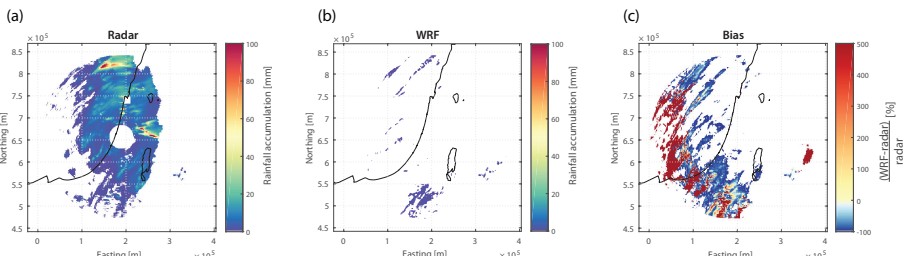

**Figure 8.** Same as Fig. 7a-c for HPE #5 (31-Mar-1993 09:00 to 02-Apr-1993 02:00; Table S1).

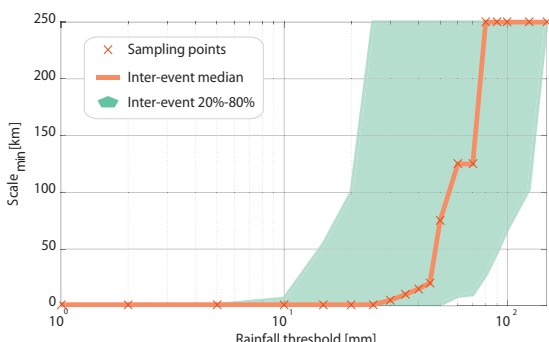

**Figure 9.** Minimal scale (see Fig. 7f and supplementary material [S1]) derived for all 41 events for various rainfall thresholds.

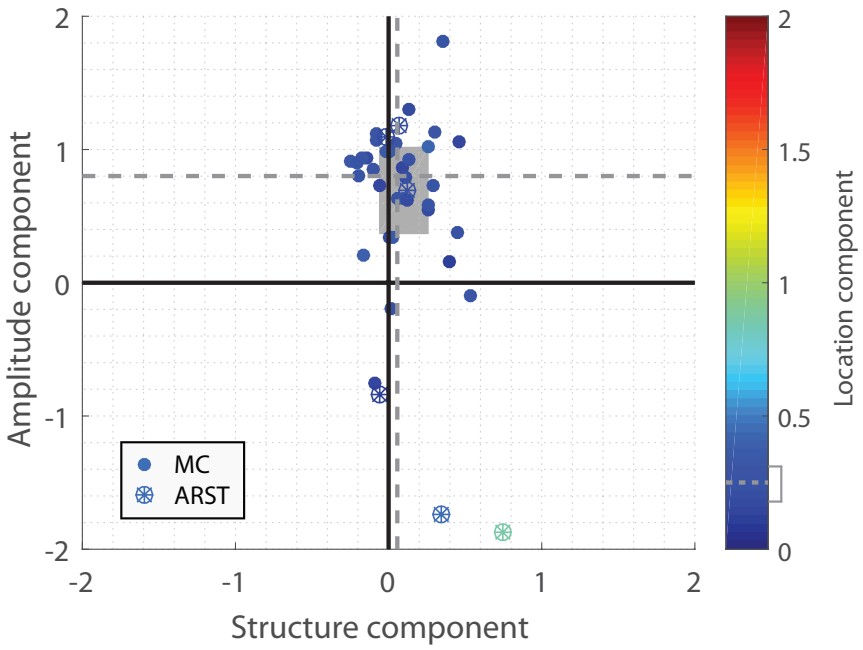

**Figure 10.** Structure-Amplitude-Location (SAL) analysis (Wernli et al., 2008). Each dot represents one event (classified according to Sect. 3.4). Dashed lines are median component values and the grey rectangle represents the $25^{th} - 75^{th}$ percentile ranges. Location component median value is 0.25, and its $25^{th} - 75^{th}$ range is 0.18–0.31. More details are in the supplementary material (S2).

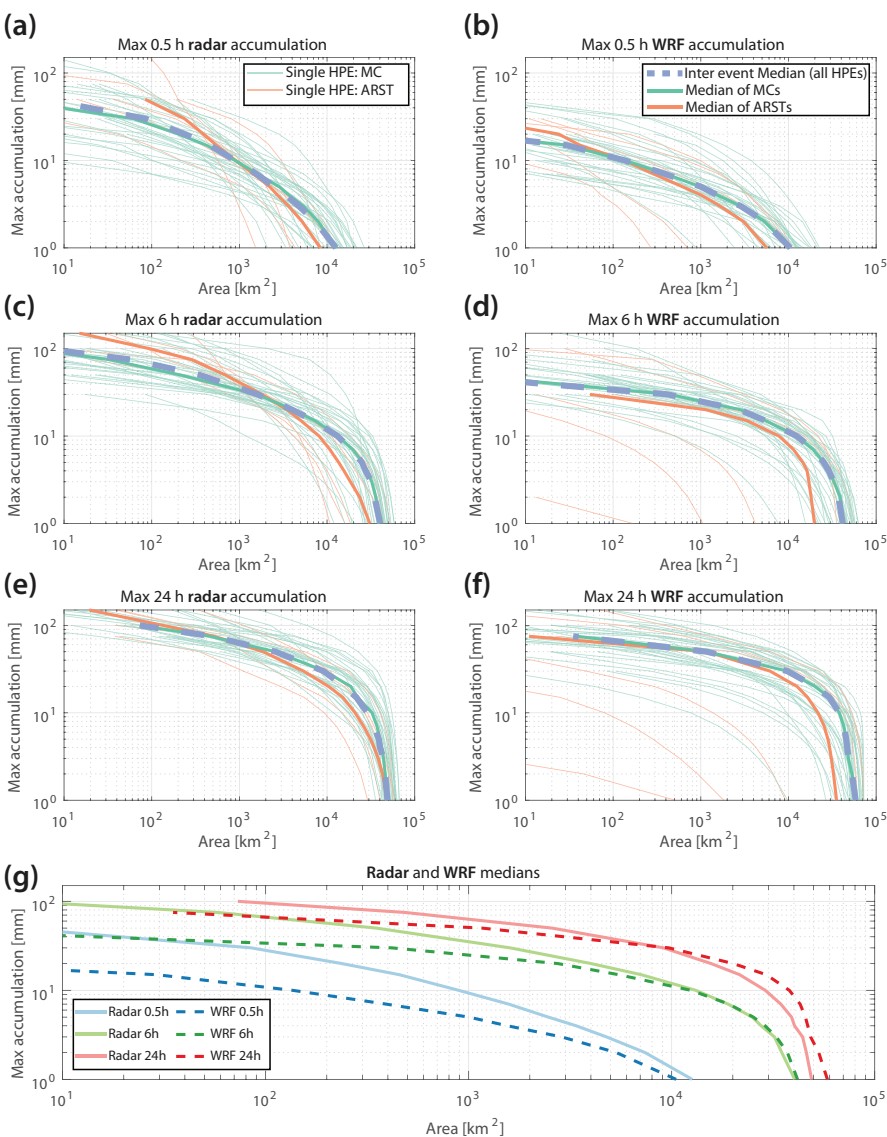

**Figure 11.** Depth-Area-Duration (DAD) curves showing the maximal amount of rainfall as a function of area, derived from the radar QPE (left; a, c and e) and from the WRF model (right; b, d and f) for 0.5 h (top), 6 h (middle) and 24 h (bottom). Green and orange lines represent HPEs classified as MCs and ARSTs, respectively. Thick lines represent the inter-event median. This median is compared between radar-QPE and WRF rainfall in panel g.

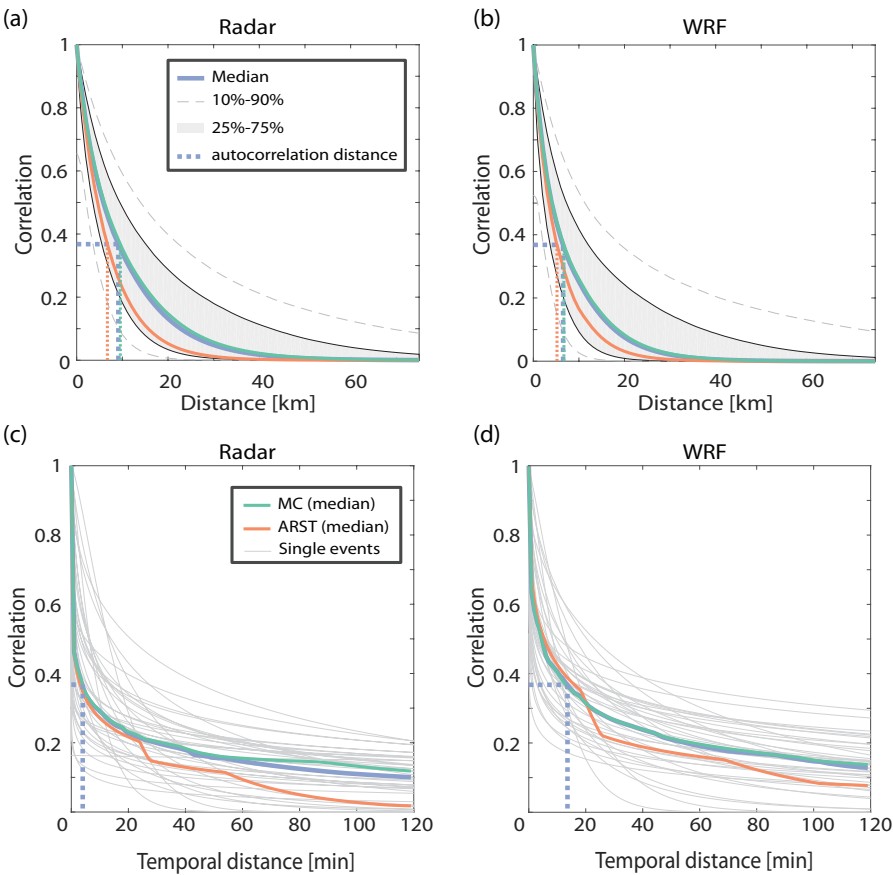

**Figure 12.** 1-D exponential fitting of rain field spatial (a, b) and temporal (c, d) autocorrelation values from radar-derived QPE (a, c) and from the WRF model (b, d). These were computed using 10-min snapshots of rain and only for periods where convective rainfall is present. Quantiles in spatial autocorrelation (a, b) represent 11,731 snapshots of radar 10-min data (10,095 of which come from MC-type events), and 14,323 WRF rainfall snapshots (12,220 of which come from MC-type events). Temporal autocorrelation plots (c, d) are composed of the 41 examined HPEs (grey), and their median values for all events (purple), for MC-type only (green) and for ARST-type only (orange).

**Table 1.** WRF model settings and specifications.

|  | Outer nest | Middle nest | Inner nest |
| --- | --- | --- | --- |
| Domains |  |  |  |
| Spatial resolution [km] | $25 \times 25$ | $5 \times 5$ | $1 \times 1$ |
| Temporal resolution [s] | $\sim 100$ | $\sim 20$ | $4 - 8$ |
| Domain size [pixels] | $100 \times 100$ | $221 \times 221$ | $551 \times 551$ |
| Number of vertical layers | 68 | 68 | 68 |
| Model top [hPa] | 25 | 25 | 25 |
| Physics |  |  |  |
| Cumulus scheme | Tiedtke (Tiedtke, 1989; Zhang et al., 2011a) |  | $-$ |
| Microphysical scheme | Thompson (Thompson et al., 2008) |  |  |
| Radiative transfer scheme | RRTMG Shortwave and Longwave (Iacono et al., 2008) |  |  |
| Planetary boundary layer scheme | Mellor–Yamada–Janjić (Janjić, 1994) |  |  |
| Surface layer scheme | Eta Similarity Scheme (Janjić, 1994) |  |  |
| Land surface model | Unified Noah Land Surface (Tewari et al., 2004) |  |  |