# Peer review of "Radar-based characterisation of heavy precipitation in the eastern Mediterranean and its representation in a convection-permitting model"

_Hydrology and Earth System Sciences, 2019_

## Referee Comment (RC1) · Anonymous Referee #1 · 30 Oct 2019

**General comments and manuscript summary:** In the submitted manuscript, the authors use 24 years of historical radar data to identify historical heavy precipitation events (HPEs) in Israel, based on various threshold criteria. These 41 HPEs are then re-simulated using the WRF model at convection-permitting resolution (1 km grid spacing). Following this, the manuscript is primarily focused on evaluating how realistically the WRF model simulates the precipitation of the 41 HPEs, compared with what the radar shows. In addition to that, the radar data are used to identify common characteristics of HPEs in the study region.

The manuscript is primarily a model evaluation study of high-resolution WRF for eastern Mediterranean HPEs, with some accompanying radar-based climatological analysis. From the scientific/technical perspective, everything seems OK. My comments which follow in the next sections are thus of a technical and minor nature, and the main question I need to answer here as a reviewer is if the paper presents sufficiently "novel concepts, ideas, tools, or data" to justify publication in HESS?

This manuscript is certainly not the first to evaluate if "the model description of rainfall during HPEs" in a convection-permitting model (CPM) is "credible", despite the claims of the authors (L62). There is even a study investigating just that with WRF in the eastern Mediterranean (Zittis et al., 2017), which surprisingly wasn't cited. For other studies asking similar questions in other regions see, for example, Berthou et al. (2018), Brisson et al. (2018), Chan et al. (2014), Chen et al. (2001), Hally et al. (2014), Kendon et al. (2012), Lean et al. (2008); many more CPM evaluation studies can be found – both event-based and climatological. This manuscript represents another contribution to this important topic. I think the publication of the manuscript can be justified on the following grounds: (1) the authors' event-based approach incorporates an unusually high number of events, which is different to the most common approaches of either continuous multi-year simulations (e.g. Ban et al., 2014) or just a handful of events (e.g. Coppola et al., 2018); (2) the authors incorporate a nice range of temporal and spatial diagnostics which are (to my knowledge) not prevalent in the extant CPM-evaluation literature, presumably because of the rarity of such long radar archives (24 years) with high spatiotemporal resolution as used by the authors; (3) CPM evaluation studies for this region of the world are not well represented in the literature.

**Specific comments:**

1. Structure of results. I wonder would the authors consider that it might make more sense to present some of the results from the characterization of rainfall patterns section (S4.2) at the start of the results section, i.e. before model biases are presented? For example, Section 4.2.1 is based on observations rather than model evaluation. It would seem more logical to me to first present the characteristics of the observed HPEs to readers and then examine if these characteristics are reproduced by the model. Indeed, in your abstract (L13-15) you present the manuscript contents in this order. However, this is for the authors to decide!

2. Title. It is not really apparent from the title of the manuscript that this is primarily a model evaluation study. I expect your results will be of most interest to readers concerned with the quality of CPM simulations, however I fear that due to the title the manuscript might be overlooked by readers searching for such information and not reach the full audience it deserves. If it was my manuscript, I'd go for a title along the lines of "Heavy precipitation in the eastern Mediterranean and its representation in a convection-permitting model". This is, of course, for the authors to decide!

3. Poorly simulated events. Of the 41 HPEs, you identify two which are simulated particularly poorly and observe that these were characterised by short storm durations (L256-257) and were highly localized (L500-501). You also suggest that the poor simulation may be due to a poorly represented moisture field in the ERA-Interim lateral boundary conditions (L466-467). Have you checked this (if possible)? It would be interesting to know if there was any trace of these precipitation events in (i) the ERA-Interim precipitation fields, or (ii) the coarser resolution WRF domains. If the boundary and initial conditions are inadequate, then there is of course no chance for WRF to well reproduce the event. But this doesn't mean that WRF itself is deficient

or is incapable of simulating such events! Maybe WRF could simulate the event using data assimilation techniques beyond the scope of this experiment, or with better boundary conditions.

4. Expectations of CPMs. My final substantive point is about what we should expect from convection-permitting models, i.e. should we expect them to match radar on a pixel-by-pixel basis? And if they can't do this, does it represent a poor simulation? This is discussed in the introduction of Roberts (2008), where it is argued that the main added value of higher-resolution precipitation forecasts should be seen in area averages – e.g. over a catchment – rather than at specific point locations. I think it's also important to remember that the observed event is also just one possible realisation of the event and WRF will never have perfect initial conditions. You correctly (L469-473) advocate the utility of ensemble simulations for HPEs in the discussion, i.e. as a means of characterizing uncertainty.

Similar information to the aforementioned could potentially additionally be presented in the introduction or during the results, as the authors see fit.

5. Data availability. I think that Section 8 about data availability is inadequate. If someone wants to reproduce your results, a bit more than the two non-specific web domains (L517-518) is needed. Is there a specific web page or ftp server where the radar and rain gauge data can be downloaded? If so, please provide the links. If not, then provide more information about how the data can be found. Additionally, what about the WRF model simulations? Will (have) you upload(ed) them to an open-access server? If so, provide the download link. Or are they available by contacting the corresponding author? Finally, I suggest uploading the WRF namelist.input as an asset when you are resubmitting the manuscript.

6. Proof reading. There are a large number of minor grammar errors throughout the text, which are too numerous to list. I therefore suggest a thorough proof reading prior to resubmission.

**Minor and technical comments:**

- Section 3.2. Could you please also state (i) the number of vertical levels and height of the model top, (ii) if shallow convection is parametrized in the inner nest, (iii) the interpolation method used, i.e. bilinear, nearest-neighbour, conservative, etc. (i) and (ii) could also be added to table 1, if appropriate.

- Figure 1. It looks like the domain boundaries have been drawn by simply finding the domain corners and drawing straight lines between them. The lower/upper boundaries of Lambert conformal domains shouldn't have constant latitudes. I think you need to extract the outermost rows/columns from WRF's XLONG and XLAT arrays and use these to plot your domain boundaries.

- Figure 2. I wonder would it make more sense to compute the %-bias? i.e. instead of bias = WRF/Radar, use bias = 100.*(WRF − Radar)/Radar. With the current formulation the dry biases are lower bounded whereas the wet biases are not upper bounded. With %-bias this would not be the case. I suppose it's not really that big of a deal. The authors can decide for themselves.

- Figure 2. Please add "a, b, c, d" labels to the panel plots, to match the text.

- L123: Note that it should be possible in WRF to just save precipitation at 10-minute intervals and other variables at a lower frequency, to reduce storage space.

- L128: I think the reference to "Sect 3.2" is wrong.

- L170: The abbreviation "TP" isn't defined anywhere.

- L396-398: It may prove difficult to identify which days to downscale from the GCMs, especially for convective events. There are some papers recently suggesting methods for identifying the best days to downscale (Chan et al., 2018; Meredith et al., 2018; Gómez-Navarro et al., 2019).

**References:**

Ban, N., Schmidli, J., Schär, C. (2014). Evaluation of the convection-resolving regional climate modeling approach in decade-long simulations. Journal of Geophysical Research: Atmospheres, 119, 7889–7907.

Berthou, S., Kendon, E. J., Chan, S. C., Ban, N., Leutwyler, D., Schär, C., Fosser, G. (2018). Pan-european climate at convection-permitting scale: A model intercomparison study. Climate Dynamics, 1–25.

Brisson, E., Brendel, C., Herzog, S., Ahrens, B. (2018). Lagrangian evaluation of convective shower characteristics in a convection-permitting model. Meteorologische Zeitschrift, 59-66.

Chan, S., Kendon, E., Fowler, H., Blenkinsop, S, Roberts, N. (2014). Projected increases in summer and winter UK sub-daily precipitation extremes from high-resolution regional climate models. Environmental Research Letters, 9(8), 084019.

Chan, S. C., Kendon, E. J., Roberts, N., Blenkinsop, S., Fowler, H. J. (2018). Large-scale predictors for extreme hourly precipitation events in convection-permitting climate simulations. Journal of Climate, 31(6), 2115-2131.

Chen, F., Warner, T. T., Manning, K. (2001). Sensitivity of orographic moist convection to landscape variability: a study of the Buffalo Creek, Colorado, flash flood case of 1996. Journal of the Atmospheric Sciences, 58(21), 3204-3223.

Coppola, E., Sobolowski, S., Pichelli, E., Raffaele, F., Ahrens, B., Anders, I., ... Caldas-

Alvarez, A. (2018). A first-of-its-kind multi-model convection permitting ensemble for investigating convective phenomena over Europe and the Mediterranean. Climate Dynamics, 1-32.

Gómez-Navarro, J. J., Raible, C. C., García-Valero, J. A., Messmer, M., Montávez, J. P., Martius, O. (2019). Event selection for dynamical downscaling: a neural network approach for physically-constrained precipitation events. Climate Dynamics, 1-17.

Meredith, E. P., Rust, H. W., Ulbrich, U. (2018). A classification algorithm for selective dynamical downscaling of precipitation extremes. Hydrology and Earth System Sciences, 22(8), 4183-4200.

Kendon, E. J., Roberts, N. M., Senior, C. A., Roberts, M. J. (2012). Realism of rainfall in a very high-resolution regional climate model. Journal of Climate, 25(17), 5791–5806.

Lean, H. W., Clark, P. A., Dixon, M., Roberts, N. M., Fitch, A., Forbes, R., Halliwell, C. (2008). Characteristics of high-resolution versions of the Met Office Unified Model for forecasting convection over the United Kingdom. Monthly Weather Review, 136(9), 3408-3424.

Roberts, N. (2008). Assessing the spatial and temporal variation in the skill of precipitation forecasts from an NWP model. Meteorological Applications: A journal of forecasting, practical applications, training techniques and modelling, 15(1), 163-169.

Zittis, G., Bruggeman, A., Camera, C., Hadjinicolaou, P., Lelieveld, J. (2017). The added value of convection permitting simulations of extreme precipitation events over the eastern Mediterranean. Atmospheric research, 191, 20-33.

---

## Referee Comment (RC2) · Anonymous Referee #2 · 3 Nov 2019

The manuscript presents a study focusing on HPEs using weather radar data and convection-permitting numerical simulations. Overall, it is an interesting study that merits publication. In particular, the consideration of a long radar data time series is important, deviating from the common practice of considering a few HPEs. Further, the methodology followed for evaluating model performance is thorough, providing useful insights. I recommend publication subject to minor revisions summarised as follows.

Comments 1. Title: I believe that the title of the manuscript is a bit misleading. To my view, the authors focus more on evaluating the WRF model at convection-permitting

[Figure]

scales, than providing a study for the characterisation of HPEs in the study region. Hence, I would suggest changing the title of the manuscript, to better reflect the real subject of the presented study. 2. Sect. 4.1.2: Two poorly simulated events were identified and some reasoning is provided in the Discussion, mainly focused on the quality of the large-scale driving reanalysis. Therefore, it would be interesting to know if the authors did check the driving ERA-Interim data for these two events, and if so, what can be concluded? Were it really an issue of bad boundary conditions? In addition, what were the results obtained from the coarser resolution domains? Were they equally poor? Such an elaboration would strengthen the authors' claim about the poor model performance. 3. Section 4.2.1 could be moved up, before the presentation of the model evaluation, as it discusses results based on observations. 4. Fig.3: Instead of presenting the WRF/RADAR ratio, the authors should consider presenting either the bias (WRF-RADAR) or transform the ratio to %. This would facilitate the interpretation of evaluation results. 5. L123: It would be useful to provide information on the interpolation method? Was it bilinear, bicubic? 6. Quality of the figures needs improvement for readability. 7. The manuscript text needs a thorough proof-reading for correcting numerous grammar and spelling errors.

---

## Referee Comment (RC3) · Anonymous Referee #3 · 14 Nov 2019

General comments:

The paper summarizes a comprehensive compilation of heavy precipitation events (HPEs) in the Eastern Mediterranean (EM) based on high-resolution radar data and WRF simulations. This set of events can be representative of the climatology in this area, and is used to quantify the spatio-temporal characteristics of HPE, and the ability to numerically predict the patterns of HPEs. A collection of four diagnostics are used to typify and contrast the radar-based and WRF spatio-temporal precipitation patterns. The events are further classified according to the synoptic situation responsible for the

[Figure]

HPEs: namely, Mediterranean cyclone (MC) and active Red Sea trough (ARST). This topic is important as it serves as a benchmark for using numerical weather prediction for flood forecasts, as well as for downscaling of future climate projections. Overall, the presentation is very good, although some excessive text can still be made easier to read, as I suggest in the following. Two major weaknesses of the results and their organization should be fully addressed before the paper can be considered for publication, as detailed in the first two Specific Comments below. Other specific comments should also be clarified.

Specific comments:

1. An important distinction is made between HPE under ARST and MC. However, the classification is not maintained throughout the results, having in mind the double goal of the paper: (i) characterize HPE patterns and (ii) evaluate WRF performance. In its current form, the classification is merely mentioned, while referring to previous works on the different spatio-temporal patterns under MC/ARST, but this is not directly shown here, with an exception of Fig. 11a-f. As mentioned in the text, HPE related to ARST is harder for prediction because of the local characteristic convection which dominates the patterns. On the other hand, HPE under MC is characterized by a cold front structure. To enhance the presentation of the results in light of the MC/ARST classification, and to make the discussion and conclusions robust, I suggest to (i) show the spatio-temporal patterns separately for each group (ii) compare the radar/WRF bias between HPE-ARST and HPE-MC. The two aspects can be achieved by reorganization of the presentation of the results, and showing figures such as Fig. 2, 3, 6, 7, 11g and 12 in light of the classification. By doing this, it will be interesting to see if there are consistent differences in the model performance, and substantiate the discussion in Lines 449-458.

2. Two individual HPE events are shown in more detail. They are important to get a better grasp of the patterns and the model/radar biases and the diagnostics used. It is, however, remaining unclear if the reader should take these results as representative,

and if so, of what. It is mentioned that HPE #1 is of MC type, while HPE #5 is ARST-type. Are they representative of the two types? Since both cases perform badly in terms of the SAL diagnostic, why do you focus on them? As the message of the work is to demonstrate the overall good performance of WRF, I find this confusing, and suggest to also illustrate the point with a case where WRF performs representatively well. I suggest to clarify this issue by explaining the rationale behind choosing to focus on these events. Further, it will make an easier reading to mark the chosen events onto Figs. 6,7,10,11.

3. Table S1 and Fig. 8: How is HPE duration calculated, and what does it mean if an HPE has a 48-h duration but no shorter durations (e.g., HPE #6)? This is confusing and should be clarified. Consequently, the results in Fig. 8 are confusing, and it is not very clear to me what we can learn from this figure.

4. Section 3.5.4 is difficult to understand, and the description of the 2D autocorrelation field, its ellipticity and orientation in Lines 379-392 is not also not clear when not referring to a figure. Please enhance or clarify these parts, possibly with an illustrative figure, such that the analysis can be standalone without referring to the references.

5. I suggest to move the spatio-temporal characteristics in Fig. 9 and 10 to earlier on in the text, even to when presenting the list of events in Sec. 3.4. This seems more natural to understand the events characteristics before assessing the model performance.

Technical corrections:

1. Line 10: add 'spatio-temporal' before 'patterns', and elaborate on what you mean by 'effects'.

2. Line 78: replace 'getting a' by 'receiving'.

3. Line 101: Add the coordinates of Ben-Gurion airport.

4. Line 130: replace 'Other' by 'Additional'.

5. Section 3.1: add more details about the radar such as: wavelength (The authors mentioned about the C band), radar parameters (reflectivity, doppler, etc). What is the maximum range of the radar observations? We see it very clearly in Fig. 1b, but number will further clarify.

6. Fig. 2: Are the white areas on the eastern side of the circle domain masked out according to the black line in Fig. 3c? If so, this should be mentioned.

7. Fig. 3: There is no legend of (a), (b), (c) and (d) as mentioned in the caption and text.

8. Figs. 3,4,5: a normalized difference (e.g., (WRF-radar)/radar) would make more sense than a ratio WRF/radar, such that the red areas will not distract the attention from more important biases.

9. Line 210: Section 3.5.3 please add a sentence to motivate the use of the DAD curve.

10. Fig. 4e: replace the scatter plot by a density plot, to see the details inside the black area.

11. Fig. 5: add the equivalent Fig. 4d-f to this case.

12. Line 165: the synoptic classification is based on semi-objective classification by Alpert (2004). This classification is based on parameters such as T, P, U and V at 1000 hPa once per day based on NCEP-NCAR reanalysis with coarse resolution ãĂŰ2.5ãĂŮˆ∘. The model (WRF) was analyzed with six hourly ERA-Interim reanalysis with 80km horizontal resolution. It is worth mentioning this.

13. Line 266: greater than 99% of pixels': do you mean to write 'corresponding to less than 1% of the pixels in this HPE'?

14. Line 278: which bias to you refer to in the square brackets? 15. Line 336: missing 'a' after 'are'.

16. Line 437: remove 'a' before 'catchments'.

17.Line 490: replace 'weather' by 'numerical weather prediction'.

18. Fig. 11a-f: make the green and blue colors more distinguishable.

19. Fig. 12: would 'temporal lag' be more suitable than 'temporal distance'?

———————————————————

---

## Author Comment (AC1) · 12 Dec 2019

**General comments and manuscript summary**: In the submitted manuscript, the authors use 24 years of historical radar data to identify historical heavy precipitation events (HPEs) in Israel, based on various threshold criteria. These 41 HPEs are then re-simulated using the WRF model at convection-permitting resolution (1 km grid spacing). Following this, the manuscript is primarily focused on evaluating how realistically the WRF model simulates the precipitation of the 41 HPEs, compared with what the radar shows. In addition to that, the radar data are used to identify common characteristics of HPEs in the study region.

The manuscript is primarily a model evaluation study of high-resolution WRF for eastern Mediterranean HPEs, with some accompanying radar-based climatological analysis. From the scientific/technical perspective, everything seems OK. My comments which follow in the next sections are thus of a technical and minor nature, and the main question I need to answer here as a reviewer is if the paper presents sufficiently "novel concepts, ideas, tools, or data" to justify publication in HESS?

  We thank reviewer #1 for acknowledging our scientific and technical work. We hope that our answers and revisions, in part proposed by reviewer #1, result in an improved contribution that justify publication in HESS. The reviewer is highly appreciated for the time and efforts dedicated for improving our manuscript. The additional references suggested by the reviewer will (a) complete the literature review, and (b) further emphasise the advances we made relative to the existing literature. In the revised manuscript we will address the issues raised by the reviewer as detailed below.

  The comments made by reviewer #1 helped us understand that we did not emphasise enough the uniqueness of the high-resolution characterisation itself, and we therefore intend to explain it better in the revised manuscript. Long, high-resolution rainfall data records (24 yr) are truly scarce, and we therefore think that this characterisation is interesting even on its own. Currently, the characterisation is detailed in section 4.2. To validate the model, each one of the pattern-related parameters we have characterised was also checked using model simulations of the same events.

This manuscript is certainly not the first to evaluate if "the model description of rainfall during HPEs" in a convection-permitting model (CPM) is "credible", despite the claims of the authors (L62). There is even a study investigating just that with WRF in the eastern Mediterranean (Zittis et al., 2017), which surprisingly wasn't cited. For other studies asking similar questions in other regions see, for example, Berthou et al. (2018), Brisson et al. (2018), Chan et al. (2014), Chen et al. (2001), Hally et al. (2014), Kendon et al. (2012), Lean et al. (2008); many more CPM evaluation studies can be found – both event-based and climatological. This manuscript represents another contribution to this important topic. I think the publication of the manuscript can be justified on the following grounds: (1) the authors' event-based approach incorporates an unusually high number of events, which is different to the most common approaches of either continuous multi-year simulations (e.g. Ban et al., 2014) or just a handful of events (e.g. Coppola et al., 2018); (2) the authors incorporate a nice range of temporal and spatial diagnostics which are (to my knowledge) not prevalent in the extant CPM-evaluation literature, presumably because

of the rarity of such long radar archives (24 years) with high spatiotemporal resolution as used by the authors; (3) CPM evaluation studies for this region of the world are not well represented in the literature.

We much appreciate the reviewer's view about our contribution. It is true that we are not the first to answer this question ("Is the model description of rainfall during HPEs credible?" [Line 62]), and we have referred in the text to many of the previous studies of the topic, but, to the best of our knowledge, we are the first to systematically do it for all the available HPEs over a 24-year period. Furthermore, there is still much to contribute in this area (in our case, we address specifically **rainfall space-time patterns** during all the available **HPEs** during a period of 24 yr). We do understand that the wording we chose may be misleading, and we would change it in the revised manuscript, so it will not be read as if we are claiming to be the first to answer this question. We actually did not know the paper you have mentioned (Zittis et al., 2017), and we are glad that you have referred us to it, since it presents a much needed conclusion both about the WRF performance during extreme rainfall events in the eastern Mediterranean and about the need for good precipitation data, even if based on a more limited number (5) of HPEs. Thus, we will refer to this paper in the revised manuscript.

**Specific comments:**

1. Structure of results. I wonder would the authors consider that it might make more sense to present some of the results from the characterization of rainfall patterns section (S4.2) at the start of the results section, i.e. before model biases are presented? For example, Section 4.2.1 is based on observations rather than model evaluation. It would seem more logical to me to first present the characteristics of the observed HPEs to readers and then examine if these characteristics are reproduced by the model. Indeed, in your abstract (L13-15) you present the manuscript contents in this order. However, this is for the authors to decide!

We understand the reviewer's point and we thought quite a lot on the best order of steps – first HPEs characteristics from radar and then model skill to reproduce those characteristics (as suggested by the reviewer) or first model skill and then HPEs characteristics as manifested in observations (radar) and model. Our tendency towards the latter approach is due to our understanding that radar observations are not perfect and have their own limitations. Therefore, we prefer to present HPE characteristics from the two sources and to emphasise both agreements and disagreements between them. This comparison follows model skill assessment. We do however agree that some of the HPEs characterisation can be moved to the first part of the results section, specifically those that are not relying on pattern analysis, i.e., seasonality and relation between HPEs at different durations (presently shown in Fig. 8 and 9). Therefore, we will make some changes in the structure of the results section: starting with general properties of HPEs, then model skill, following by space-time HPEs characteristics detected from observations and model. Accordingly, we will make small modifications in the abstract and the introduction sections.

E.g., in the introduction (line 71) we will replace the word "Then" by "Although our observations are based on radar data, these are certainly not perfect. Therefore, we quantified several…"

2. Title. It is not really apparent from the title of the manuscript that this is primarily a model evaluation study. I expect your results will be of most interest to readers concerned with the quality of CPM simulations, however I fear that due to the title the manuscript might be overlooked by readers searching for such information and not reach the full audience it deserves. If it was my manuscript, I'd go for a title along the lines of "Heavy precipitation in the eastern Mediterranean and its representation in a convection-permitting model". This is, of course, for the authors to decide!

We agree, but we do want to keep the "characterisation" part, from the reasons stated above. We suggest to change the title to:
"Radar-based characterisation of heavy precipitation in the eastern Mediterranean and its representation in a convection-permitting model".

3. Poorly simulated events. Of the 41 HPEs, you identify two which are simulated particularly poorly and observe that these were characterised by short storm durations (L256-257) and were highly localized (L500-501). You also suggest that the poor simulation may be due to a poorly represented moisture field in the ERA-Interim lateral boundary conditions (L466-467). Have you checked this (if possible)? It would be interesting to know if there was any trace of these precipitation events in (i) the ERA-Interim precipitation fields, or (ii) the coarser resolution WRF domains. If the boundary and initial conditions are inadequate, then there is of course no chance for WRF to well reproduce the event. But this doesn't mean that WRF itself is deficient or is incapable of simulating such events! Maybe WRF could simulate the event using data assimilation techniques beyond the scope of this experiment, or with better boundary conditions.

We agree with the suggestion. We will show (in the supplementary materials) the results of the coarsest WRF domain. This could possibly give an idea of both the WRF simulated rain fields and of the ERA-Interim input. To have an impression of it, we attach below a preliminary analysis of the rainfall for the first of these two events (event #5; 31/3/93-2/4/93), based on the WRF coarsest domain, to be compared with Fig 5. In contrast to most of the simulated HPEs, in which rainfall was simulated quite well in the innermost WRF domain, this event had almost no rainfall simulated in the inner domain. As the figure below shows, rainfall was not produced by the WRF coarsest domain over the area where it was observed (Fig 5), but rather a few hundred km from there – suggesting that the initial conditions were insufficient to produce rainfall in vicinity of the observed one, regardless to the spatial error of the small-scale (innermost) domain. As the reviewer states, it might have been better simulated using data assimilation, or any other better boundary conditions. However, both are beyond the scope of our manuscript.

[Figure]

Figure: Rainfall in the coarsest WRF domain during HPE #5 (Table S1) and the approximate range of the Shacham radar (Figure 1).

4. Expectations of CPMs.

My final substantive point is about what we should expect from convection-permitting models, i.e. should we expect them to match radar on a pixel-by-pixel basis? And if they can't do this, does it represent a poor simulation? This is discussed in the introduction of Roberts (2008), where it is argued that the main added value of higher-resolution precipitation forecasts should be seen in area averages – e.g. over a catchment – rather than at specific point locations. I think it's also important to remember that the observed event is also just one possible realisation of the event and WRF will never have perfect initial conditions. You correctly (L469-473) advocate the utility of ensemble simulations for HPEs in the discussion, i.e. as a means of characterizing uncertainty. Similar information to the aforementioned could potentially additionally be presented in the introduction or during the results, as the authors see fit.

The point raised by the reviewer is a crucial one that we want to stress out in the manuscript, and it is actually one of the main points we examine in this manuscript. This is the reason we utilise neighbourhood-based rainfall pattern measures (SAL, FSS), rather than pixel-based indices of success (Fig. 4d, 4f, 6, 7 versus Fig 4e). Moreover, when we compare

rainfall patterns, we consider the centre-of-mass of precipitation, Depth-Area-Duration (DAD) curves, and spatial and temporal autocorrelation curves, all of which are not based on point observations. We will better stress this aspect in the revised manuscript. Specifically, we plan to add to the discussion (line 465) the following: "The main added value of convection-permitting models is seen in area averages, rather than over small scale regions (Roberts, 2008). Therefore, over large catchments (e.g., larger than a few hundreds of square kilometres, as suggested by the minimal scale presented in Fig. 6) their forecasts should be relatively useful and accurate. Nonetheless, the use of such a deterministic model is still unsatisfactory in…".

5. Data availability. I think that Section 8 about data availability is inadequate. If someone wants to reproduce your results, a bit more than the two non-specific web domains (L517-518) is needed. Is there a specific web page or ftp server where the radar and rain gauge data can be downloaded? If so, please provide the links. If not, then provide more information about how the data can be found. Additionally, what about the WRF model simulations? Will (have) you upload(ed) them to an openaccess server? If so, provide the download link. Or are they available by contacting the corresponding author? Finally, I suggest uploading the WRF namelist.input as an asset when you are resubmitting the manuscript.

We agree with this comment, however not all of the data are owned by us or can be publicly accessed. We suggest to add to the revised version of the manuscript the specific domain from which one can download the rain gauge data (https://ims.data.gov.il/). These data are not ours to give, however it is available through this data archive (unfortunately, only in Hebrew). The radar data are also not ours to give. It was provided to us by "EMS-Mekorot projects". However, if needed, corrected and gauge-adjusted data (previously published in (Marra and Morin, 2015)) could be given, in the form of images, through a personal communication with the head of the Hydrometeorology lab in the Hebrew University of Jerusalem, Prof. Efrat Morin (efrat.morin@mail.huji.ac.il).

The size of the simulation results is really big (~4.6 TB), so we prefer not to upload those results to the web. We accept your suggestion, and we will add the WRF namelist.input file to the supplementary materials. Using the namelist and the ERA-Interim input files, one will be able to fully reproduce our results.

6. Proof reading. There are a large number of minor grammar errors throughout the text, which are too numerous to list. I therefore suggest a thorough proof reading prior to resubmission.

Accepted. We will proof read the manuscript thoroughly, once we revise it.

**Minor and technical comments:**

- Section 3.2. Could you please also state (i) the number of vertical levels and height of the model top, (ii) if shallow convection is parametrized in the inner nest, (iii) the interpolation method used, i.e. bilinear, nearest-neighbour, conservative, etc. (i) and (ii) could also be added to table 1, if appropriate.

(i) The number of vertical levels is 68, as stated in Table 1 and the top of the model is at 25 hPa., and we will state this information in the revised version of the manuscript (ii) We use the WRF Tiedtke scheme in the two outer domains (as stated in Table 1) that has a shallow cumulus component, as detailed in (Tiedtke, 1989; Zhang et al., 2011). We will detail this part in the text, as it seems not to be clear from Table 1 only. (iii) The interpolation method used is simply nearest-neighbour, and we will write it in the revised version of the manuscript. Moreover, as suggested, we intend to add the WRF namelist files, so all of the details of our simulations will be clearer.

- Figure 1. It looks like the domain boundaries have been drawn by simply finding the domain corners and drawing straight lines between them. The lower/upper boundaries of Lambert conformal domains shouldn't have constant latitudes. I think you need to extract the outermost rows/columns from WRF's XLONG and XLAT arrays and use these to plot your domain boundaries.

That's correct. The domains are not plotted with their exact extent. We will correct this in the revised version of the manuscript.

- Figure 2. I wonder would it make more sense to compute the %-bias? i.e. instead of bias = WRF/Radar, use bias = 100.*(WRF – Radar)/Radar. With the current formulation the dry biases are lower bounded whereas the wet biases are not upper bounded. With %-bias this would not be the case. I suppose it's not really that big of a deal. The authors can decide for themselves.

This was also mentioned by the other reviewers. We will change the bias definition into normalised difference (i.e. (WRF-radar)/radar).

- Figure 2. Please add "a, b, c, d" labels to the panel plots, to match the text.

We accept this correction, and we will apply it in the revised version of the manuscript.

- L123: Note that it should be possible in WRF to just save precipitation at 10-minute intervals and other variables at a lower frequency, to reduce storage space.

That's correct. Still, after doing this (we actually saved few 2D variables, and not only the precipitation field, however we did not save 3D fields every 10-min), because of the high resolution, the results weigh on average ~112 GB per event.

- L128: I think the reference to "Sect 3.2" is wrong.

Right. This should be corrected to "Sect 3.3" and we intend to add a reference to Table S1 as well.

- L170: The abbreviation "TP" isn't defined anywhere

> Correct. We will change this abbreviation to the full synoptic class name (i.e. "Tropical Plume").

- L396-398: It may prove difficult to identify which days to downscale from the GCMs, especially for convective events. There are some papers recently suggesting methods for identifying the best days to downscale (Chan et al., 2018; Meredith et al., 2018; Gómez-Navarro et al., 2019).

> This is true, and we can refer to these papers if requested. However, recent studies also tried running long-term downscaled simulations (e.g., Kendon et al., 2014; Liu et al., 2017).

**References from reviewer #1:**

Ban, N., Schmidli, J., Schär, C. (2014). Evaluation of the convection-resolving regional climate modeling approach in decade-long simulations. Journal of Geophysical Research: Atmospheres, 119, 7889–7907.

Berthou, S., Kendon, E. J., Chan, S. C., Ban, N., Leutwyler, D., Schär, C., Fosser, G. (2018). Pan-european climate at convection-permitting scale: A model intercomparison study. Climate Dynamics, 1–25.

Brisson, E., Brendel, C., Herzog, S., Ahrens, B. (2018). Lagrangian evaluation of convective shower characteristics in a convection-permitting model. Meteorologische Zeitschrift, 59-66.

Chan, S., Kendon, E., Fowler, H., Blenkinsop, S, Roberts, N. (2014). Projected increases in summer and winter UK sub-daily precipitation extremes from high-resolution regional climate models. Environmental Research Letters, 9(8), 084019.

Chan, S. C., Kendon, E. J., Roberts, N., Blenkinsop, S., Fowler, H. J. (2018). Largescale predictors for extreme hourly precipitation events in convection-permitting climate simulations. Journal of Climate, 31(6), 2115-2131.

Chen, F., Warner, T. T., Manning, K. (2001). Sensitivity of orographic moist convection to landscape variability: a study of the Buffalo Creek, Colorado, flash flood case of 1996. Journal of the Atmospheric Sciences, 58(21), 3204-3223.

Coppola, E., Sobolowski, S., Pichelli, E., Raffaele, F., Ahrens, B., Anders, I., ... Caldas-Alvarez, A. (2018). A first-of-its-kind multi-model convection permitting ensemble for investigating convective phenomena over Europe and the Mediterranean. Climate Dynamics, 1-32.

Gómez-Navarro, J. J., Raible, C. C., García-Valero, J. A., Messmer, M., Montávez, J. P., Martius, O. (2019). Event selection for dynamical downscaling: a neural network approach for physically-constrained precipitation events. Climate Dynamics, 1-17.

Meredith, E. P., Rust, H. W., Ulbrich, U. (2018). A classification algorithm for selective dynamical downscaling of precipitation extremes. Hydrology and Earth System Sciences, 22(8), 4183-4200.

Kendon, E. J., Roberts, N. M., Senior, C. A., Roberts, M. J. (2012). Realism of rainfall in a very high-resolution regional climate model. Journal of Climate, 25(17), 5791– 5806.

Lean, H. W., Clark, P. A., Dixon, M., Roberts, N. M., Fitch, A., Forbes, R., Halliwell, C. (2008). Characteristics of high-resolution versions of the Met Office Unified Model for forecasting convection over the United Kingdom. Monthly Weather Review, 136(9), 3408-3424.

Roberts, N. (2008). Assessing the spatial and temporal variation in the skill of precipitation forecasts from an NWP model. Meteorological Applications: A journal of forecasting, practical applications, training techniques and modelling, 15(1), 163-169.

Zittis, G., Bruggeman, A., Camera, C., Hadjinicolaou, P., Lelieveld, J. (2017). The added value of convection permitting simulations of extreme precipitation events over the eastern Mediterranean. Atmospheric research, 191, 20-33.

References:

Kendon, E. J., Roberts, N. M., Fowler, H. J., Roberts, M. J., Chan, S. C. and Senior, C. A.: Heavier summer downpours with climate change revealed by weather forecast resolution model, Nat. Clim. Chang., 4(7), 570–576, doi:10.1038/nclimate2258, 2014.

Liu, C., Ikeda, K., Rasmussen, R., Barlage, M., Newman, A. J., Prein, A. F., Chen, F., Chen, L., Clark, M., Dai, A., Dudhia, J., Eidhammer, T., Gochis, D., Gutmann, E., Kurkute, S., Li, Y., Thompson, G. and Yates, D.: Continental-scale convection-permitting modeling of the current and future climate of North America, Clim. Dyn., 49(1–2), 71–95, doi:10.1007/s00382-016-3327-9, 2017.

Marra, F. and Morin, E.: Use of radar QPE for the derivation of Intensity–Duration–Frequency curves in a range of climatic regimes, J. Hydrol., 531, 427–440, doi:10.1016/j.jhydrol.2015.08.064, 2015.

Tiedtke, M.: A Comprehensive Mass Flux Scheme for Cumulus Parameterization in Large-Scale Models, Mon. Weather Rev., 117(8), 1779–1800, doi:10.1175/1520-0493(1989)117<1779:ACMFSF>2.0.CO;2, 1989.

Zhang, C., Wang, Y. and Hamilton, K.: Improved Representation of Boundary Layer Clouds over the Southeast Pacific in ARW-WRF Using a Modified Tiedtke Cumulus Parameterization Scheme, Mon. Weather Rev., 139(11), 3489–3513, doi:10.1175/MWR-D-10-05091.1, 2011.

Zittis, G., Bruggeman, A., Camera, C., Hadjinicolaou, P. and Lelieveld, J.: The added value of convection permitting simulations of extreme precipitation events over the eastern Mediterranean, Atmos. Res., 191, 20–33, doi:10.1016/j.atmosres.2017.03.002, 2017.

---

## Author Comment (AC2) · 12 Dec 2019

The manuscript presents a study focusing on HPEs using weather radar data and convection-permitting numerical simulations. Overall, it is an interesting study that merits publication. In particular, the consideration of a long radar data time series is important, deviating from the common practice of considering a few HPEs. Further, the methodology followed for evaluating model performance is thorough, providing useful insights. I recommend publication subject to minor revisions summarised as follows.

We highly appreciate the reviewer's comments regarding our manuscript. We will address all of the comments raised, as detailed below, in the revised version of the manuscript.

Comments

1. Title: I believe that the title of the manuscript is a bit misleading. To my view, the authors focus more on evaluating the WRF model at convection-permitting scales, than providing a study for the characterisation of HPEs in the study region. Hence, I would suggest changing the title of the manuscript, to better reflect the real subject of the presented study.

This idea was also raised by reviewer #1. We understand that we did not focus enough attention to our presentation of HPEs' characterisation, which is unique due to the high resolution rainfall data and the relatively large number of events. We intend to better emphasise this part of the paper in the revised manuscript. Moreover, to better present our paper, we plan to substitute the current title with the following one: "Radar-based characterisation of heavy precipitation in the eastern Mediterranean and its representation in a convection-permitting model".

2. Sect. 4.1.2: Two poorly simulated events were identified and some reasoning is provided in the Discussion, mainly focused on the quality of the large-scale driving reanalysis. Therefore, it would be interesting to know if the authors did check the driving ERA-Interim data for these two events, and if so, what can be concluded? Were it really an issue of bad boundary conditions? In addition, what were the results obtained from the coarser resolution domains? Were they equally poor? Such an elaboration would strengthen the authors' claim about the poor model performance.

This is a good point, also raised by reviewer #1. To address it, we plan to include some analyses of the coarser domains in the WRF simulations. It is hard to tell for sure if the boundary conditions are bad, because we do not have better data than the reanalysis, and comparison of other reanalyses or data assimilation techniques are beyond the scope of this manuscript. However, in contrast to most of the model simulations, in which rainfall was quite well simulated, in these two HPEs the innermost domain exhibited almost no rain. At the coarsest model domain, there is no rainfall simulated over the region, but rather only hundreds of km from the observed rain (compare the figure below with Fig 5). Below is a preliminary analysis of rainfall for one of the two events (event #5 in table S1).

[Figure]

Figure: Rainfall in the coarsest WRF domain during HPE #5 (Table S1) and the approximate range of the Shacham radar (Figure 1).

3. Section 4.2.1 could be moved up, before the presentation of the model evaluation, as it discusses results based on observations.

We understand the reviewer's point and we thought quite a lot on the right order – first HPEs characteristics from radar and then model skill to reproduce those characteristics (as suggested by the reviewer) or first model skill and then HPEs characteristics as manifested in observations (radar) and model. Our tendency towards the latter approach is due to our understanding that radar observations are not perfect and have their own limitations. Therefore, we prefer to present HPEs characteristics from the two sources and to emphasise both agreements and disagreements between them. This comparison follows model skill assessment. We do however agree that some of the HPEs characterisation can be moved to the first part of the result section, specifically those that are not relying on pattern analysis, i.e., seasonality and relation between HPEs at different durations (presently shown in Fig. 8 and 9). Therefore, we will make some changes in the structure of the results section: starting with general properties of HPEs, then model skill, following by space-time HPEs

characteristics detected from observations and model. Accordingly, we will make small modifications in the abstract section.

4. Fig.3: Instead of presenting the WRF/RADAR ratio, the authors should consider presenting either the bias (WRF-RADAR) or transform the ratio to %. This would facilitate the interpretation of evaluation results.

Agreed. We will change our bias definition into normalised difference (i.e. (WRF-radar)/radar).

5. L123: It would be useful to provide information on the interpolation method? Was it bilinear, bicubic?

The interpolation method we used is simply nearest-neighbour, and we will write it in the revised version of the manuscript.

6. Quality of the figures needs improvement for readability.

Thank you for this comment. We will review our figures and make them clearer if needed for the revised manuscript. Moreover, we plan to upload the final figures in a vectorised format (wherever possible), so that their quality in any case would be improved.

7. The manuscript text needs a thorough proof-reading for correcting numerous grammar and spelling errors.

Accepted. We will proof read the manuscript thoroughly.

---

## Author Comment (AC3) · 12 Dec 2019

General comments:

The paper summarizes a comprehensive compilation of heavy precipitation events (HPEs) in the Eastern Mediterranean (EM) based on high-resolution radar data and WRF simulations. This set of events can be representative of the climatology in this area, and is used to quantify the spatio-temporal characteristics of HPE, and the ability to numerically predict the patterns of HPEs. A collection of four diagnostics are used to typify and contrast the radar-based and WRF spatio-temporal precipitation patterns. The events are further classified according to the synoptic situation responsible for the HPEs: namely, Mediterranean cyclone (MC) and active Red Sea trough (ARST). This topic is important as it serves as a benchmark for using numerical weather prediction for flood forecasts, as well as for downscaling of future climate projections. Overall, the presentation is very good, although some excessive text can still be made easier to read, as I suggest in the following. Two major weaknesses of the results and their organization should be fully addressed before the paper can be considered for publication, as detailed in the first two Specific Comments below. Other specific comments should also be clarified.

We thank the reviewer for the comments provided, and for the time and efforts spent in reviewing the manuscript. We have carefully addressed all of the comments and we believe the revised manuscript will benefit from them.

Specific comments:

1. An important distinction is made between HPE under ARST and MC. However, the classification is not maintained throughout the results, having in mind the double goal of the paper: (i) characterize HPE patterns and (ii) evaluate WRF performance. In its current form, the classification is merely mentioned, while referring to previous works on the different spatio-temporal patterns under MC/ARST, but this is not directly shown here, with an exception of Fig. 11a-f. As mentioned in the text, HPE related to ARST is harder for prediction because of the local characteristic convection which dominates the patterns. On the other hand, HPE under MC is characterized by a cold front structure. To enhance the presentation of the results in light of the MC/ARST classification, and to make the discussion and conclusions robust, I suggest to (i) show the spatio-temporal patterns separately for each group (ii) compare the radar/WRF bias between HPE-ARST and HPE-MC. The two aspects can be achieved by reorganization of the presentation of the results, and showing figures such as Fig. 2, 3, 6, 7, 11g and 12 in light of the classification. By doing this, it will be interesting to see if there are consistent differences in the model performance, and substantiate the discussion in Lines 449-458.

We thank the reviewer for raising this point. We agree the distinction between ARST and MC is important both for HPE pattern characterisation and for the ability to forecast the events with a NWP model. Accordingly, we will modify some of the figures, as detailed below, to present this distinction, and further discuss it in the discussion section.

Fig. 2 & 3: HPE identification is based on specific rainfall thresholds that do not take the classification into account but rather the local quantiles (Sect 3.3). We do not think these thresholds should be defined with classification, since it will reduce their robustness. In addition, the distinction between regions that are better observed by the radar (Fig. 3) would

not benefit from synoptic classification. Therefore, synoptic distinction is not relevant for Figs. 2 and 3.

Fig. 6: Although, in principle, the FSS median and range shown in Fig. 6 for all HPEs can be computed for each synoptic type separately, it should be noted that we deal with only 6 ARST-type HPEs, out of them, two are not well simulated. Since we cannot provide a reliable statistic for ARST type we would not include in Fig. 6 the distinction between the two types. We do plan however to refer in the text to some general differences, qualitatively identified, from FSS analysis of the individual HPEs for each type.

Fig 7: We will add the synoptic distinction to the SAL analysis presented in Fig. 7 and in the text discussing these results.

Fig. 10: We will add the synoptic distinction to the figure.

Fig. 11: The DAD analysis is already classified into the two types of synoptic circulation patterns, however, to make a better distinction between rain-fields based on their duration of accumulation and their source (radar-QPE / WRF), we extracted the median curves from each one of the sub figures (a-f) and enlarged them in panel g. We feel that adding a synoptic distinction to this panel, may attract the attention of the reader from the distinction between durations and the source of the rainfall, which was the purpose of panel g.

Fig. 12: We will add the synoptic distinction to the figure and discuss its results.

2. Two individual HPE events are shown in more detail. They are important to get a better grasp of the patterns and the model/radar biases and the diagnostics used. It is, however, remaining unclear if the reader should take these results as representative, and if so, of what. It is mentioned that HPE #1 is of MC type, while HPE #5 is ARSTtype. Are they representative of the two types? Since both cases perform badly in terms of the SAL diagnostic, why do you focus on them? As the message of the work is to demonstrate the overall good performance of WRF, I find this confusing, and suggest to also illustrate the point with a case where WRF performs representatively well. I suggest to clarify this issue by explaining the rationale behind choosing to focus on these events. Further, it will make an easier reading to mark the chosen events onto Figs. 6,7,10,11.

The two events shown in Figs. 4, 5 are meant to represent one well-simulated event (event #1, shown in Fig. 4) and one poorly-simulated event (#5, Fig. 5). It seems there was a confusion with the two poorly simulated events (#5 and #20, Table S1) discussed later on, but this was not the intension; we will modify the text to better clarify this point (see below). The two events shown in Figs. 4 and 5 are given as an example to show what the model is able (Fig. 4) or unable (Fig. 5) to simulate, and they also exemplify a typical MC and a typical ARST cases. It turns out that it is harder for the model to represent the localised rainfall that often happens during ARSTs. The different performance for the two cases is very clear from the SAL analysis (see figure below).

[Figure]

Figure: SAL analysis of the 41 HPEs. MC-type of HPEs denoted with circles and ARST-type with triangles.

In light of your comment, we will better clarify in the revised manuscript the purpose of the closer look at these two events (Section 4.12): "…localised rainfall. Fig. 4a-c presents as an example a well-simulated HPE case (event #1, Table S1). In addition, the distributions of rainfall among pixels is generally well represented (Fig. 4d). At the same time, pixel-based comparisons are deemed inappropriate for such an analysis, as shown in the scatter plot (Fig 4e). Most of the examined HPEs led to similar observations with the exception of two HPEs in which the WRF model clearly failed in representing the rainfall patterns. An example for such a poor simulation is given on Fig. 5 (event #5, Table S1)".

3. Table S1 and Fig. 8: How is HPE duration calculated, and what does it mean if an HPE has a 48-h duration but no shorter durations (e.g., HPE #6)? This is confusing and should be clarified. Consequently, the results in Fig. 8 are confusing, and it is not very clear to me what we can learn from this figure.

The term "event duration" in Fig. 8, Table S1 and possibly in other sections of the paper, does not refer to the total duration of the event but rather to the duration according to which it was selected as HPE. We defined HPEs, in Sect 3.3 by "the exceedance of local, quantile-based thresholds over a sufficiently large area… For a set of durations between 1 and 72 hours we defined the threshold as the 99.5th quantile of the non-zero (i.e. >0.1 mm) hourly amounts observed in each radar pixel… we classify as HPEs all time intervals during which at least 1000 pixels (i.e., 1000 km2) exceeded their local threshold". This is to say that, if enough pixels in the radar archive exceeded their own threshold, for a given examined duration, we defined this event as an HPE for this duration. Obviously, a given event can be selected for several examined durations. Fig. 8 shows that it is hard to separate events according to their duration, i.e. the duration for which rain intensity was exceeded the threshold, because of the above overlapping. However, we see that it is not clear enough. We will clarify it in the text, and possibly move Fig 8. earlier, to relate also to the details in Sect. 3.3.

4. Section 3.5.4 is difficult to understand, and the description of the 2D autocorrelation field, its ellipticity and orientation in Lines 379-392 is not also not clear when not referring to a figure. Please enhance or clarify these parts, possibly with an illustrative figure, such that the analysis can be standalone without referring to the references.

We agree and make sure the description is clear enough by its own in the revised manuscript. However, we do not want to add a lot of text to describe what was already published (e.g., in Marra and Morin, 2018). Therefore, we would try to describe better than we did, but still briefly.

5. I suggest to move the spatio-temporal characteristics in Fig. 9 and 10 to earlier on in the text, even to when presenting the list of events in Sec. 3.4. This seems more natural to understand the events characteristics before assessing the model performance.

We agree that it seems more natural to talk about the characteristics of events before presenting the model performance. However, our goal is to characterise HPEs and to evaluate model capability in reproducing those characteristics. Therefore, we chose to present the examined characteristics from both radar and model, and thus these analyses come after model performance results. However, accounting for suggestions from all reviewers, we would like the re-order the results section as follows:
   a) General properties of HPEs (presently shown e.g., in Fig. 8 and 9)
   b) Model skill (Fig 3-7)
   c) Comparison of characteristics between the radar-QPE and the model (Fig. 10-12)

Technical corrections:

1. Line 10: add 'spatio-temporal' before 'patterns', and elaborate on what you mean by 'effects'.

Accepted. We will change this sentence to "Spatiotemporal rainfall patterns govern the hydrological, geomorphological and societal effects of HPEs".

2. Line 78: replace 'getting a' by 'receiving'.

Accepted.

3. Line 101: Add the coordinates of Ben-Gurion airport.

Agreed. We will add "31.998N, 34.908E".

4. Line 130: replace 'Other' by 'Additional'.

OK.

5. Section 3.1: add more details about the radar such as: wavelength (The authors mentioned about the C band), radar parameters (reflectivity, doppler, etc). What is the maximum range of the radar observations? We see it very clearly in Fig. 1b, but number will further clarify.

We will add the radar wavelength (5.35 cm), its range (185 km), the fact that it is a non-doppler radar, and that raw radar reflectivity data were translated to QPE using first a fixed Z-R relationship, $Z = 316 \cdot R^{1.5}$ , and then into QPE by applying physically based corrections and gauge-based adjustment procedures (see details in Marra and Morin(2015)).

6. Fig. 2: Are the white areas on the eastern side of the circle domain masked out according to the black line in Fig. 3c? If so, this should be mentioned.

Yes. OK, we would add this to the figure caption.

7. Fig. 3: There is no legend of (a), (b), (c) and (d) as mentioned in the caption and text.

Thanks for noting this. We will add the legend to the figure.

8. Figs. 3,4,5: a normalized difference (e.g., (WRF-radar)/radar) would make more sense than a ratio WRF/radar, such that the red areas will not distract the attention from more important biases.

Accepted. We will change our definition for the bias to be a normalized difference (i.e. ((WRF-radar)/radar)).

9. Line 210: Section 3.5.3 please add a sentence to motivate the use of the DAD curve.

We agree that a motivation is needed. Therefore, section 3.5.3 starts with a motivation sentence ("Areal rainfall amounts are crucial drivers of the hydrologic response and are important for understanding rainfall structure and triggering mechanisms"). We think that this text is enough for the aims of this manuscript.

10. Fig. 4e: replace the scatter plot by a density plot, to see the details inside the black area.

Accepted. We will replace it.

11. Fig. 5: add the equivalent Fig. 4d-f to this case.

Panels d-f were added to Fig. 4 to show that even in a well-simulated event, there is a large disagreement on a pixel scale, while the general characteristics (considering all pixels, as in the histogram, or a large neighbourhood, as in the FSS analysis) could still be well simulated. Fig. 5, however, presents a poorly-simulated event and it is not much successful no matter in what perspective we examine it. This is why we did not present further analyses of rainfall patterns. We also think that due to the clarification made in response to the reviewer's 2nd comment this point is now clearer.

12. Line 165: the synoptic classification is based on semi-objective classification by Alpert (2004). This classification is based on parameters such as T, P, U and V at 1000 hPa once per day based on NCEP-NCAR reanalysis with coarse resolution ãAˇU2.5ã ˝ AˇUˆ˚◦. The model (WRF) was analyzed with six hourly ERA-Interim reanalysis with 80km horizontal resolution. It is worth mentioning this.

Agreed. We will mention it.

13. Line 266: greater than 99% of pixels': do you mean to write 'corresponding to less than 1% of the pixels in this HPE'?

Yes. This was our intention, and we will edit the text accordingly.

14. Line 278: which bias to you refer to in the square brackets?

It is the bias of the median (inter-event) amplitude score (from the SAL analysis). However, we will explain this better in the revised text.

15. Line 336: missing 'a' after 'are'.

Correct. Thank you for noticing.

16. Line 437: remove 'a' before 'catchments'.

Accepted.

17. Line 490: replace 'weather' by 'numerical weather prediction'.

Accepted.

18. Fig. 11a-f: make the green and blue colors more distinguishable.

OK.

19. Fig. 12: would 'temporal lag' be more suitable than 'temporal distance'?

It is suitable, however we tried to follow the term used in Marra and Morin (2018) that refer to time-distance.

References:

Marra, F. and Morin, E.: Use of radar QPE for the derivation of Intensity–Duration–Frequency curves in a range of climatic regimes, J. Hydrol., 531, 427–440, doi:10.1016/j.jhydrol.2015.08.064, 2015.

Marra, F. and Morin, E.: Autocorrelation structure of convective rainfall in semiarid-arid climate derived from high-resolution X-Band radar estimates, Atmos. Res., 200(September 2017), 126–138, doi:10.1016/j.atmosres.2017.09.020, 2018.

---

## Author Response (AR1)

**Dear Editor,**

We were glad to read the positive reviews and the interest shown in our research, and we are grateful for the timely and useful reviews provided. Please find below the detailed response (blue font) to all comments made by the reviewers, in accordance to our responses in the interactive discussion.

Following the comments made by the reviewers we rewrote some segments of the text as shown in the two "track changes" texts below (one for the main text and the other for the supplementary) and detailed here.

The manuscript and supplementary files were uploaded separately in their revised form.

Sincerely,

Moshe Armon, on behalf of all authors.

**Anonymous Referee #1**

**General comments and manuscript summary**: In the submitted manuscript, the authors use 24 years of historical radar data to identify historical heavy precipitation events (HPEs) in Israel, based on various threshold criteria. These 41 HPEs are then re-simulated using the WRF model at convection-permitting resolution (1 km grid spacing). Following this, the manuscript is primarily focused on evaluating how realistically the WRF model simulates the precipitation of the 41 HPEs, compared with what the radar shows. In addition to that, the radar data are used to identify common characteristics of HPEs in the study region.

The manuscript is primarily a model evaluation study of high-resolution WRF for eastern Mediterranean HPEs, with some accompanying radar-based climatological analysis. From the scientific/technical perspective, everything seems OK. My comments which follow in the next sections are thus of a technical and minor nature, and the main question I need to answer here as a reviewer is if the paper presents sufficiently "novel concepts, ideas, tools, or data" to justify publication in HESS?

We thank reviewer #1 for acknowledging our scientific and technical work. We hope that our answers and revisions, in part proposed by reviewer #1, result in an improved contribution that justify publication in HESS. The reviewer is highly appreciated for the time and efforts dedicated for improving our manuscript. The additional references suggested by the reviewer will (a) complete the literature review, and (b) further emphasise the advances we made relative to the existing literature. In the revised manuscript we will address the issues raised by the reviewer as detailed below.

The comments made by reviewer #1 helped us understand that we did not emphasise enough the uniqueness of the high-resolution characterisation itself, and we therefore explain it better in the revised manuscript (Lines 75-77). Long, high-resolution rainfall data records (24 yr) are truly scarce, and we therefore think that this characterisation is interesting even on its own. Currently, the characterisation is detailed in section 4.4. To validate the model, each one of the pattern-related parameters we have characterised was also checked using model simulations of the same events.

This manuscript is certainly not the first to evaluate if "the model description of rainfall during HPEs" in a convection-permitting model (CPM) is "credible", despite the claims of the authors (L62). There is even a study investigating just that with WRF in the eastern Mediterranean (Zittis et al., 2017), which surprisingly wasn't cited. For other studies asking similar questions in other regions see, for example, Berthou et al. (2018), Brisson et al. (2018), Chan et al. (2014), Chen et al. (2001), Hally et al. (2014), Kendon et al. (2012), Lean et al. (2008); many more CPM evaluation studies can be found – both event-based and climatological. This manuscript represents another contribution to this important topic. I think the publication of the manuscript can be justified on the following grounds: (1) the authors' event-based approach incorporates an unusually high number of events, which is different to the most common approaches of either continuous multi-year simulations (e.g. Ban et al., 2014) or just a handful of events (e.g. Coppola et al., 2018); (2) the authors incorporate a nice range of temporal and spatial diagnostics which are (to my knowledge) not prevalent in the extant CPM-evaluation literature, presumably because

of the rarity of such long radar archives (24 years) with high spatiotemporal resolution as used by the authors; (3) CPM evaluation studies for this region of the world are not well represented in the literature.

We much appreciate the reviewer's view about our contribution. It is true that we are not the first to answer this question ("Is the model description of rainfall during HPEs credible?" [Line 62]), and we have referred in the text to many of the previous studies of the topic, but, to the best of our knowledge, we are the first to systematically do it for all the available HPEs over a 24-year period. Furthermore, there is still much to contribute in this area (in our case, we address specifically **rainfall space-time patterns** during all the available **HPEs** during a period of 24 yr). We do understand that the wording we chose may be misleading, thus we changed it in the revised manuscript, so it will not be read as if we are claiming to be the first to answer this question (L65). We actually did not know the paper you have mentioned (Zittis et al., 2017), and we are glad that you have referred us to it, since it presents a much needed conclusion both about the WRF performance during extreme rainfall events in the eastern Mediterranean and about the need for good precipitation data, even if based on a more limited number (5) of HPEs. Thus, we referred to this paper in the revised manuscript (L58-60).

**Specific comments:**

1. Structure of results. I wonder would the authors consider that it might make more sense to present some of the results from the characterization of rainfall patterns section (S4.2) at the start of the results section, i.e. before model biases are presented? For example, Section 4.2.1 is based on observations rather than model evaluation. It would seem more logical to me to first present the characteristics of the observed HPEs to readers and then examine if these characteristics are reproduced by the model. Indeed, in your abstract (L13-15) you present the manuscript contents in this order. However, this is for the authors to decide!

We understand the reviewer's point and we thought quite a lot on the best order of steps – first HPEs characteristics from radar and then model skill to reproduce those characteristics (as suggested by the reviewer) or first model skill and then HPEs characteristics as manifested in observations (radar) and model. Our tendency towards the latter approach is due to our understanding that radar observations are not perfect and have their own limitations. Therefore, we prefer to present HPE characteristics from the two sources and to emphasise both agreements and disagreements between them. This comparison follows model skill assessment. We do however agree that some of the HPEs characterisation can be moved to the first part of the results section, specifically those that are not relying on pattern analysis, i.e., seasonality and relation between HPEs at different durations (previously shown in Fig. 8 and 9). Therefore, we made some changes in the structure of the results section: starting with general properties of HPEs (presently in Section 4.1), then model skill (4.2-4.3), following by space-time HPEs characteristics

detected from observations and model (4.4). Accordingly, we made small modifications in the introduction section. E.g., in the introduction (line 77) we replaced the word "Then" by "Considering that our observations are based on radar data, they are certainly not perfect. Therefore, we quantified and compared ..."

2. Title. It is not really apparent from the title of the manuscript that this is primarily a model evaluation study. I expect your results will be of most interest to readers concerned with the quality of CPM simulations, however I fear that due to the title the manuscript might be overlooked by readers searching for such information and not reach the full audience it deserves. If it was my manuscript, I'd go for a title along the lines of "Heavy precipitation in the eastern Mediterranean and its representation in a convection-permitting model". This is, of course, for the authors to decide!

We agree, but we do want to keep the "characterisation" part, from the reasons stated above. We changed the title to:

"Radar-based characterisation of heavy precipitation in the eastern Mediterranean and its representation in a convection-permitting model".

3. Poorly simulated events. Of the 41 HPEs, you identify two which are simulated particularly poorly and observe that these were characterised by short storm durations (L256-257) and were highly localized (L500-501). You also suggest that the poor simulation may be due to a poorly represented moisture field in the ERA-Interim lateral boundary conditions (L466-467). Have you checked this (if possible)? It would be interesting to know if there was any trace of these precipitation events in (i) the ERA-Interim precipitation fields, or (ii) the coarser resolution WRF domains. If the boundary and initial conditions are inadequate, then there is of course no chance for WRF to well reproduce the event. But this doesn't mean that WRF itself is deficient or is incapable of simulating such events! Maybe WRF could simulate the event using data assimilation techniques beyond the scope of this experiment, or with better boundary conditions.

We agree with the suggestion. We presently show (in the supplementary materials) the results of the coarsest WRF domain. This could possibly give an idea of both the WRF simulated rain fields and of the ERA-Interim input. To have an impression of it, we attach below a preliminary analysis of the rainfall for the first of these two events (event #5; 31/3/93-2/4/93), based on the WRF coarsest domain, to be compared with Fig 8. In contrast to most of the simulated HPEs, in which rainfall was simulated quite well in the innermost WRF domain, this event had almost no rainfall simulated in the inner domain. As the figure below shows, rainfall was not produced by the WRF coarsest domain over the area where it was observed (Fig 8), but rather a few hundred km from there – suggesting that the initial conditions were insufficient to produce rainfall in vicinity of the observed one, regardless to the spatial error of the small-scale (innermost) domain. As the reviewer states, it might have been better simulated using data assimilation, or any other better boundary conditions. However, both are beyond the scope of our manuscript.

Figure: Rainfall in the coarsest WRF domain during HPE #5 (Table S1) and the approximate range of the Shacham radar (Figure 1).

**4. Expectations of CPMs.**

My final substantive point is about what we should expect from convection-permitting models, i.e. should we expect them to match radar on a pixel-by-pixel basis? And if they can't do this, does it represent a poor simulation? This is discussed in the introduction of Roberts (2008), where it is argued that the main added value of higher-resolution precipitation forecasts should be seen in area averages – e.g. over a catchment – rather than at specific point locations. I think it's also important to remember that the observed event is also just one possible realisation of the event and WRF will never have perfect initial conditions. You correctly (L469-473) advocate the utility of ensemble simulations for HPEs in the discussion, i.e. as a means of characterizing uncertainty. Similar information to the aforementioned could potentially additionally be presented in the introduction or during the results, as the authors see fit.

The point raised by the reviewer is a crucial one that we want to stress out in the manuscript, and it is actually one of the main points we examine in this manuscript. This is the reason we utilise neighbourhood-based rainfall pattern measures (SAL, FSS), rather than pixel-based indices of success (Fig. 7d, 7f, 9, 10 versus Fig 7e). Moreover, when we compare

rainfall patterns, we consider the centre-of-mass of precipitation, Depth-Area-Duration (DAD) curves, and spatial and temporal autocorrelation curves, all of which are not based on point observations. We will better stress this aspect in the revised manuscript. Specifically, we added to the discussion (lines 493-496) the following: "The main added value of convection-permitting models is seen in area averages, rather than over small-scale regions (Roberts, 2008). Therefore, over large catchments (e.g., larger than a few hundred square kilometres, as suggested by the minimal scale presented in Fig. 9), their forecasts are expected to be relatively useful and accurate. Nonetheless, the use of a deterministic convection-permitting model is still unsatisfactory for pinpointing the highest observed rain accumulations...".

5. Data availability. I think that Section 8 about data availability is inadequate. If someone wants to reproduce your results, a bit more than the two non-specific web domains (L517-518) is needed. Is there a specific web page or ftp server where the radar and rain gauge data can be downloaded? If so, please provide the links. If not, then provide more information about how the data can be found. Additionally, what about the WRF model simulations? Will (have) you upload(ed) them to an openaccess server? If so, provide the download link. Or are they available by contacting the corresponding author? Finally, I suggest uploading the WRF namelist.input as an asset when you are resubmitting the manuscript.

We agree with this comment, however not all of the data are owned by us or can be publicly accessed. We added to the revised version of the manuscript the specific domain from which one can download the rain gauge data (https://ims.data.gov.il/). These data are not ours to give, however it is available through this data archive (unfortunately, only in Hebrew). The radar data are also not ours to give. It was provided to us by "EMS-Mekorot projects". However, if needed, corrected and gauge-adjusted data (previously published in (Marra and Morin, 2015)) could be given, in the form of images, through a personal communication with the head of the Hydrometeorology lab in the Hebrew University of Jerusalem, Prof. Efrat Morin (efrat.morin@mail.huji.ac.il).

The size of the simulation results is really big ( $\sim$ 4.6 TB), so we prefer not to upload those results to the web. We accept your suggestion, and we added the WRF namelist.input file to the supplementary materials. Using the namelist and the ERA-Interim input files, one will be able to fully reproduce our results.

6. Proof reading. There are a large number of minor grammar errors throughout the text, which are too numerous to list. I therefore suggest a thorough proof reading prior to resubmission.

Accepted. We proof read the manuscript thoroughly.

Minor and technical comments:

- Section 3.2. Could you please also state (i) the number of vertical levels and height of the model top, (ii) if shallow convection is parametrized in the inner nest, (iii) the interpolation method used, i.e. bilinear, nearest-neighbour, conservative, etc. (i) and (ii) could also be added to table 1, if appropriate.

(i) The number of vertical levels is 68, as stated in Table 1 and the top of the model is at 25 hPa., this information is now shown in the manuscript (L129-130) (ii) We use the WRF Tiedtke scheme in the two outer domains (as stated in Table 1) that has a shallow cumulus component, as detailed in (Tiedtke, 1989; Zhang et al., 2011). We detailed this part in the text (L135-136), as it seems not to be clear from Table 1 only. (iii) The interpolation method used is simply nearest-neighbour, and it is now stated clearly in the text (L133). Moreover, as suggested, we intend to add the WRF namelist files, so all of the details of our simulations will be clearer.

- Figure 1. It looks like the domain boundaries have been drawn by simply finding the domain corners and drawing straight lines between them. The lower/upper boundaries of Lambert conformal domains shouldn't have constant latitudes. I think you need to extract the outermost rows/columns from WRF's XLONG and XLAT arrays and use these to plot your domain boundaries.

That's correct. The domains are not plotted with their exact extent. We have corrected this in the revised Figure 1.

- Figure 2. I wonder would it make more sense to compute the %-bias? i.e. instead of bias = WRF/Radar, use bias = 100.\*(WRF – Radar)/Radar. With the current formulation the dry biases are lower bounded whereas the wet biases are not upper bounded. With %-bias this would not be the case. I suppose it's not really that big of a deal. The authors can decide for themselves.

This was also mentioned by the other reviewers. We have changed the bias definition into normalised difference (i.e. (WRF-radar)/radar) (Sect. 4.2, Fig. 6c, 7c, 8c).

- Figure 2. Please add "a, b, c, d" labels to the panel plots, to match the text.

We accept this correction, probably intended for Figure 3, and we have applied it in the revised version of the now-named Figure 6.

- L123: Note that it should be possible in WRF to just save precipitation at 10-minute intervals and other variables at a lower frequency, to reduce storage space.

That's correct. Still, after doing this (we actually saved few 2D variables, and not only the precipitation field, however we did not save 3D fields every 10-min), because of the high resolution, the results weigh on average  $\sim$ 112 GB per event.

- L128: I think the reference to "Sect 3.2" is wrong.

Right. This is now corrected to "Sect 3.3" and we added a reference to Table S1 as well (L139).

- L170: The abbreviation "TP" isn't defined anywhere

Correct. We have changed this abbreviation to the full synoptic class name (i.e. "Tropical Plume"; L183 & L185).

- L396-398: It may prove difficult to identify which days to downscale from the GCMs, especially for convective events. There are some papers recently suggesting methods for identifying the best days to downscale (Chan et al., 2018; Meredith et al., 2018; Gómez-Navarro et al., 2019).

This is true, and we now address it in L421-422.

**References:**

Ban, N., Schmidli, J., Schär, C. (2014). Evaluation of the convection-resolving regional climate modeling approach in decade-long simulations. Journal of Geophysical Research: Atmospheres, 119, 7889–7907.

Berthou, S., Kendon, E. J., Chan, S. C., Ban, N., Leutwyler, D., Schär, C., Fosser, G. (2018). Pan-european climate at convection-permitting scale: A model intercomparison study. Climate Dynamics, 1–25.

Brisson, E., Brendel, C., Herzog, S., Ahrens, B. (2018). Lagrangian evaluation of convective shower characteristics in a convection-permitting model. Meteorologische Zeitschrift, 59-66.

Chan, S., Kendon, E., Fowler, H., Blenkinsop, S, Roberts, N. (2014). Projected increases in summer and winter UK sub-daily precipitation extremes from high-resolution regional climate models. Environmental Research Letters, 9(8), 084019.

Chan, S. C., Kendon, E. J., Roberts, N., Blenkinsop, S., Fowler, H. J. (2018). Largescale predictors for extreme hourly precipitation events in convection-permitting climate simulations. Journal of Climate, 31(6), 2115-2131.

Chen, F., Warner, T. T., Manning, K. (2001). Sensitivity of orographic moist convection to landscape variability: a study of the Buffalo Creek, Colorado, flash flood case of 1996. Journal of the Atmospheric Sciences, 58(21), 3204-3223.

Coppola, E., Sobolowski, S., Pichelli, E., Raffaele, F., Ahrens, B., Anders, I., ... Caldas-Alvarez, A. (2018). A first-of-its-kind multi-model convection permitting ensemble for investigating convective phenomena over Europe and the Mediterranean. Climate Dynamics, 1-32.

Gómez-Navarro, J. J., Raible, C. C., García-Valero, J. A., Messmer, M., Montávez, J. P., Martius, O. (2019). Event selection for dynamical downscaling: a neural network approach for physically-constrained precipitation events. Climate Dynamics, 1-17.

Meredith, E. P., Rust, H. W., Ulbrich, U. (2018). A classification algorithm for selective dynamical downscaling of precipitation extremes. Hydrology and Earth System Sciences, 22(8), 4183-4200.

Kendon, E. J., Roberts, N. M., Senior, C. A., Roberts, M. J. (2012). Realism of rainfall in a very high-resolution regional climate model. Journal of Climate, 25(17), 5791–5806.

Lean, H. W., Clark, P. A., Dixon, M., Roberts, N. M., Fitch, A., Forbes, R., Halliwell, C. (2008). Characteristics of high-resolution versions of the Met Office Unified Model for forecasting convection over the United Kingdom. Monthly Weather Review, 136(9), 3408-3424.

Roberts, N. (2008). Assessing the spatial and temporal variation in the skill of precipitation forecasts from an NWP model. Meteorological Applications: A journal of forecasting, practical applications, training techniques and modelling, 15(1), 163-169.

Zittis, G., Bruggeman, A., Camera, C., Hadjinicolaou, P., Lelieveld, J. (2017). The added value of convection permitting simulations of extreme precipitation events over the eastern Mediterranean. Atmospheric research, 191, 20-33.

**Anonymous Referee #2**

The manuscript presents a study focusing on HPEs using weather radar data and convection-permitting numerical simulations. Overall, it is an interesting study that merits publication. In particular, the consideration of a long radar data time series is important, deviating from the common practice of considering a few HPEs. Further, the methodology followed for evaluating model performance is thorough, providing useful insights. I recommend publication subject to minor revisions summarised as follows.

We highly appreciate the reviewer's comments regarding our manuscript. We have addressed all of the comments raised, as detailed below, in the revised version of the manuscript.

**Comments**

1. Title: I believe that the title of the manuscript is a bit misleading. To my view, the authors focus more on evaluating the WRF model at convection-permitting scales, than providing a study for the characterisation of HPEs in the study region. Hence, I would suggest changing the title of the manuscript, to better reflect the real subject of the presented study.

This idea was also raised by reviewer #1. We understand that we did not focus enough attention to our presentation of HPEs' characterisation, which is unique due to the high resolution rainfall data and the relatively large number of events. We better emphasised this part of the paper in the revised manuscript (L65). Moreover, to better present our paper, we have substituted the previous title with the following one: "Radar-based characterisation of heavy precipitation in the eastern Mediterranean and its representation in a convection-permitting model".

2. Sect. 4.1.2: Two poorly simulated events were identified and some reasoning is provided in the Discussion, mainly focused on the quality of the large-scale driving reanalysis. Therefore, it would be interesting to know if the authors did check the driving ERA-Interim data for these two events, and if so, what can be concluded? Were it really an issue of bad boundary conditions? In addition, what were the results obtained from the coarser resolution domains? Were they equally poor? Such an elaboration would strengthen the authors' claim about the poor model performance.

This is a good point, also raised by reviewer #1. To address it, we included some analyses of the coarser domains in the WRF simulations. It is hard to tell for sure if the boundary conditions are bad, because we do not have better data than the reanalysis, and comparison of other reanalyses or data assimilation techniques are beyond the scope of this manuscript. However, in contrast to most of the model simulations, in which rainfall was quite well simulated, in these two HPEs the innermost domain exhibited almost no rain. At the coarsest model domain, there is no rainfall simulated over the region, but rather only hundreds of km from the observed rain (compare the figure below with Fig 8). Below is a preliminary analysis of rainfall for one of the two events (event #5 in table S1).

Figure: Rainfall in the coarsest WRF domain during HPE #5 (Table S1) and the approximate range of the Shacham radar (Figure 1).

3. Section 4.2.1 could be moved up, before the presentation of the model evaluation, as it discusses results based on observations.

We understand the reviewer's point and we thought quite a lot on the right order – first HPEs characteristics from radar and then model skill to reproduce those characteristics (as suggested by the reviewer) or first model skill and then HPEs characteristics as manifested in observations (radar) and model. Our tendency towards the latter approach is due to our understanding that radar observations are not perfect and have their own limitations. Therefore, we prefer to present HPEs characteristics from the two sources and to emphasise both agreements and disagreements between them. This comparison follows model skill assessment. We do however agree that some of the HPEs characterisation can be moved to the first part of the result section, specifically those that are not relying on pattern analysis, i.e., seasonality and relation between HPEs at different durations (previously shown in Fig. 8 and 9). Therefore, we made some changes in the structure of the results section: starting

with general properties of HPEs (Sect. 4.1), then model skill (4.2-4.3), following by spacetime HPEs characteristics detected from observations and model (4.4).

4. Fig.3: Instead of presenting the WRF/RADAR ratio, the authors should consider presenting either the bias (WRF-RADAR) or transform the ratio to %. This would facilitate the interpretation of evaluation results.

Agreed. We have changed our bias definition into normalised difference (i.e. (WRF-radar)/radar) (Sect. 4.2).

5. L123: It would be useful to provide information on the interpolation method? Was it bilinear, bicubic?

The interpolation method we used is simply nearest-neighbour, and it is now stated clearly (L133).

6. Quality of the figures needs improvement for readability.

Thank you for this comment. We reviewed our figures and made them clearer for the revised manuscript. Moreover, we plan to upload the final figures in a vectorised format (wherever possible), so that their quality in any case would be improved.

7. The manuscript text needs a thorough proof-reading for correcting numerous grammar and spelling errors.

Accepted. We proof read the manuscript thoroughly.

**Anonymous Referee #3**

**General comments:**

The paper summarizes a comprehensive compilation of heavy precipitation events (HPEs) in the Eastern Mediterranean (EM) based on high-resolution radar data and WRF simulations. This set of events can be representative of the climatology in this area, and is used to quantify the spatio-temporal characteristics of HPE, and the ability to numerically predict the patterns of HPEs. A collection of four diagnostics are used to typify and contrast the radar-based and WRF spatio-temporal precipitation patterns. The events are further classified according to the synoptic situation responsible for the HPEs: namely, Mediterranean cyclone (MC) and active Red Sea trough (ARST). This topic is important as it serves as a benchmark for using numerical weather prediction for flood forecasts, as well as for downscaling of future climate projections. Overall, the presentation is very good, although some excessive text can still be made easier to read, as I suggest in the following. Two major weaknesses of the results and their organization should be fully addressed before the paper can be considered for publication, as detailed in the first two Specific Comments below. Other specific comments should also be clarified.

We thank the reviewer for the comments provided, and for the time and efforts spent in reviewing the manuscript. We have carefully addressed all of the comments and we believe the revised manuscript will benefit from them.

**Specific comments:**

1. An important distinction is made between HPE under ARST and MC. However, the classification is not maintained throughout the results, having in mind the double goal of the paper: (i) characterize HPE patterns and (ii) evaluate WRF performance. In its current form, the classification is merely mentioned, while referring to previous works on the different spatio-temporal patterns under MC/ARST, but this is not directly shown here, with an exception of Fig. 11a-f. As mentioned in the text, HPE related to ARST is harder for prediction because of the local characteristic convection which dominates the patterns. On the other hand, HPE under MC is characterized by a cold front structure. To enhance the presentation of the results in light of the MC/ARST classification, and to make the discussion and conclusions robust, I suggest to (i) show the spatio-temporal patterns separately for each group (ii) compare the radar/WRF bias between HPE-ARST and HPE-MC. The two aspects can be achieved by reorganization of the presentation of the results, and showing figures such as Fig. 2, 3, 6, 7, 11g and 12 in light of the classification. By doing this, it will be interesting to see if there are consistent differences in the model performance, and substantiate the discussion in Lines 449-458.

We thank the reviewer for raising this point. We agree the distinction between ARST and MC is important both for HPE pattern characterisation and for the ability to forecast the events with a NWP model. Accordingly, we modified some of the figures, as detailed below, to present this distinction, and further detailed it in the results section (L253, L328-330, L342-345, and discussion sections (L419) and in the abstract (L16-17).

Previously Fig. 2 & 3 (presently Fig. 2 & 6): HPE identification is based on specific rainfall thresholds that do not take the classification into account but rather the local quantiles (Sect 3.3). We do not think these thresholds should be defined with classification, since it will

reduce their robustness. In addition, the distinction between regions that are better observed by the radar (Fig. 6) would not benefit from synoptic classification. Therefore, synoptic distinction is not relevant for Figs. 2 and 6.

Previously Fig. 6 (presently Fig. 9): Although, in principle, the FSS median and range shown in Fig. 9 for all HPEs can be computed for each synoptic type separately, it should be noted that we deal with only 6 ARST-type HPEs, out of them, two are not well simulated. Since we cannot provide a reliable statistic for ARST type we would not include in Fig. 9 the distinction between the two types. We still, however, referred in the text to some general differences, qualitatively identified, from FSS analysis of the individual HPEs for each type (L328-330).

Previously Fig 7 (presently Fig. 10): We have added the synoptic distinction to the SAL analysis presented in Fig. 10 and in the text discussing these results (L342-345).

Previously Fig. 10 (presently Fig. 4): We have added the synoptic distinction to the figure. Fig. 11: The DAD analysis is already classified into the two types of synoptic circulation patterns, however, to make a better distinction between rain-fields based on their duration of accumulation and their source (radar-QPE / WRF), we extracted the median curves from each one of the sub figures (a-f) and enlarged them in panel g. We feel that adding a synoptic distinction to this panel, may attract the attention of the reader from the distinction between durations and the source of the rainfall, which was the purpose of panel g.

Fig. 12: We have added the synoptic distinction to the figure and discuss its results (Sect. 4.4.2).

2. Two individual HPE events are shown in more detail. They are important to get a better grasp of the patterns and the model/radar biases and the diagnostics used. It is, however, remaining unclear if the reader should take these results as representative, and if so, of what. It is mentioned that HPE #1 is of MC type, while HPE #5 is ARSTtype. Are they representative of the two types? Since both cases perform badly in terms of the SAL diagnostic, why do you focus on them? As the message of the work is to demonstrate the overall good performance of WRF, I find this confusing, and suggest to also illustrate the point with a case where WRF performs representatively well. I suggest to clarify this issue by explaining the rationale behind choosing to focus on these events. Further, it will make an easier reading to mark the chosen events onto Figs. 6,7,10,11.

The two events shown previously in Figs. 4, 5 (Presently in Figs. 7, 8) are meant to represent one well-simulated event (event #1, shown in Fig. 7) and one poorly-simulated event (#5, Fig. 8). It seems there was a confusion with the two poorly simulated events (#5 and #20, Table S1) discussed later on, but this was not the intension; we have modified the text to better clarify this point (see below). The two events shown in Figs. 7 and 8 are given as an example to show what the model is able (Fig. 7) or unable (Fig. 8) to simulate, and they also exemplify a typical MC and a typical ARST cases. It turns out that it is harder for the model to represent the localised rainfall that often happens during ARSTs. The different performance for the two cases is very clear from the SAL analysis (see figure below).

---

## Editor Decision (ED1)

**Editor decision**

Journal: HESS
Title: Radar-based characterisation of heavy precipitation in the eastern Mediterranean and its representation in a convection-permitting model

Dear authors,

Thanks for submitting the revised version of your manuscript. It addresses most reviewer comments adequately and the quality of English writing and grammar has much improved.
At this point, I have a couple of minor comments that I would like you to address, mainly for clarification of your statements:

Terminology:
- Duration: The way you use the term duration creates confusion (as also mentioned by a reviewer) and even with the added explanation I'm not entirely sure whether you duration refers to "duration above threshold added up over an event" or "aggregation level", i.e. observation averaged over time windows of 1 to 72 hours.
  Please provide a definition to avoid any confusion.
  The term is also used in the abstract, so please make sure its meaning can be unequivocally understood here, too.
- Characterisation: Two of the reviewers mentioned they see the paper as a model evaluation study, with accompanying climatology characterization.
  I agree with the reviewers, in that the paper does not "systematically characterise high-resolution rainfall patterns" as suggested in the Introduction. Rather, it provides a general characterization based on DAD curves and autocorrelation structure, summarized across all events.
  Please clarify this in the Introduction and Abstract.

Results section:
- In 4.3 you discuss FSS scores (p11, l 315-319): from figure 7f it's clear that the scores for 100mm and 125mm are very unstable due to a limited number of observations in these classes. No conclusions can be drawn from such unstable scores – please rephrase.
- In 4.3. p11, l 326 you mention estimation of "minimum scales for skillful forecasts for various cumulative rainfall depths". FSS is a binary score, that evaluates rainfall detection, not rainfall depths.
  Please be more precise in your phrasing and make sure to use terms like "detection" or "occurrence" instead of "rainfall depth" when referring to FSS scores (throughout section 4.3!).
- The discussion of SAL scores is quite limited, it seems there is much more room for discussion, especially of the Structure and Location scores.
  On p 11, L334: "The structure component was well modelled in most cases, showing the ability of the WRF to accurately generate precipitation objects". Can you elaborate a bit more for the reader: were objects well represented in terms of position and/or intensity, can you say anything about what explains the performance range (certain types of events that typically perform better/worse)?
  For Location (p11, L338-344): it is stated that locations of precipitation objects are not well represented, while the structure score seemed to suggest they were. Again, a bit more discussion is needed here.
- The ellipticity of the 2d autocorrelation fields are discussed as a mean across all events. While in reality ellipticity would be expected to vary a lot between events (as indicated by

the wide range of ellipticity values). Please justify why you think comparing means across events is useful?
- Discussion of performance differences: throughout section 4 (Results) you tend to attribute difference in performance between radar and WRF always to radar. You main observations at larger distance are subject to range degradation (L278, 296, 380), those at shorter distance are attributed to clutter (L372-373). In L397-398 you mention that "because radar QPE suffers from temporal inconsistencies".
This way of discussing performance differences (basically attributing all deviations to radar) is unbalanced and I'm not convinced it's correct. Unless your radar product is particularly poor, but if that's the case, then what's left of the value of this "unique dataset"?
Please check your performance evaluation in section 4 and make sure to have a more balanced discussion between the radar observations and WRF model results.

Discussion section:
- The first paragraph of the Discussion section has a lot of repetition of what's already reported in section 4. If it's meant to be a summary, it better fits at the end of section 4.
- L445: "hither and tither" should be hither and thither. Still, it's an unusual term in professional manuscripts, so I suggest to rephrase it
- Section 5.2: a discussion on the usefulness of data in relation to flash flood comes out of the blue here and has no connection with anything in earlier parts of the manuscript. It's almost a lost piece of literature review. Consider to either move it up to Section 1 where literature is discussed or remove it entirely.
- Section 5.3: please clarify whether 2nd paragraph (starting L484) still refers to ARST (as in first paragraph) or whether this is about all HPEs?

Conclusions:
- You present a nice conclusion about the minimum scale that could be used for forecasting rainfall depths in relation to threshold intensities. It would be valuable nice to directly add some numbers here that came out for your analyses, to make the conclusion more quantitative.

Figures:
- In Figure 7f a vertical scale seems to be missing, for the FSS scores?
- In Figure 11 vertical axis title seems incorrect : should be max accumulation instead of Rain threshold?

---

## Author Response (AR2)

**Dear editor,**

We would like to thank you the time and efforts spent and for your useful suggestions about our manuscript, which helped us to improve it. We have addressed all of your comments (in black), and our detailed answers are below (in blue).

Sincerely, Moshe Armon, On behalf of all authors.

**Dear authors,**

Thanks for submitting the revised version of your manuscript. It addresses most reviewer comments adequately and the quality of English writing and grammar has much improved.

At this point, I have a couple of minor comments that I would like you to address, mainly for clarification of your statements:

**Terminology:**

- Duration: The way you use the term duration creates confusion (as also mentioned by a reviewer) and even with the added explanation I'm not entirely sure whether you duration refers to "duration above threshold added up over an event" or "aggregation level", i.e. observation averaged over time windows of 1 to 72 hours. Please provide a definition to avoid any confusion. The term is also used in the abstract, so please make sure its meaning can be unequivocally understood here, too.

Accepted. In the revised manuscript we made a clear separation: the term "duration" is used for the time window for which the rain intensity is averaged, and the term "time-span" is used when addressing the time interval over which a storm occurred. We corrected the manuscript accordingly.

- Characterisation: Two of the reviewers mentioned they see the paper as a model evaluation study, with accompanying climatology characterization. I agree with the reviewers, in that the paper does not "systematically characterise high-resolution rainfall patterns" as suggested in the Introduction. Rather, it provides a general characterization based on DAD curves and autocorrelation structure, summarized across all events. Please clarify this in the Introduction and Abstract.

We understand the point of view of both the reviewers and the editor. We do however think that identifying characteristics of 41 HPEs (over 24-year record), selected in a systematic way, justifying the use of the term "characterisation". We also want to clarify that more characteristics of HPEs are included, beyond DAD curves and autocorrelation structure, specifically: (a) seasonality, (b) intensity of rain that produces HPE (in accordance to our definition of HPEs), (c) centre of mass location, and (d) distribution and co-occurrence over various examined durations. This series of characteristics, however, is presented in the first result section (4.1. Quasi-climatology of HPEs), while the DAD and autocorrelation characteristics are presented in section 4.4. We made this

change of order from the original one to address other comments of the reviewers in the previous round. Thus, we still want to leave in the sentence stating we are characterising rainfall patterns, both in the introduction and in the abstract. However, to be more precise, we changed the sentence in the introduction mentioned by the editor to: "characterise high-resolution rainfall patterns (seasonality, spatial distribution of intensities, location, and spatiotemporal structure) during HPEs in the hydroclimatically heterogeneous EM" (L73-74).

**Results section:**

- In 4.3 you discuss FSS scores (p11, I 315-319): from figure 7f it's clear that the scores for 100mm and 125mm are very unstable due to a limited number of observations in these classes. No conclusions can be drawn from such unstable scores – please rephrase.

We agree and thank the editor for pointing it out. We have rephrased the sentences: "For larger cumulative rainfall the FSS is unstable due to a limited amount of observations of such cases, and thus, no conclusions about the goodness of the results can be made for the occurrence of such high cumulative rainfall. It is only for the higher rainfall amounts, e.g., 125 mm, corresponding to less than 1% of the pixels in this HPE, that the model forecast is unskilled at all spatial scales, i.e., the uniform FSS outperforms the WRF forecast FSS (yet, these results are based on a limited amount of data as well)." (L321-326).

- In 4.3. p11, I 326 you mention estimation of "minimum scales for skillful forecasts for various cumulative rainfall depths". FSS is a binary score, that evaluates rainfall detection, not rainfall depths. Please be more precise in your phrasing and make sure to use terms like "detection" or "occurrence" instead of "rainfall depth" when referring to FSS scores (throughout section 4.3!).

This is correct, however one can still calculate the detection of cumulative rainfall exceeding an arbitrary amount. Therefore, we rephrased the paragraph in section 4.3 to be more precise (L321-326, L332-335).

- The discussion of SAL scores is quite limited, it seems there is much more room for discussion, especially of the Structure and Location scores. On p 11, L334: "The structure component was well modelled in most cases, showing the ability of the WRF to accurately generate precipitation objects". Can you elaborate a bit more for the reader: were objects well represented in terms of position and/or intensity, can you say anything about what explains the performance range (certain types of events that typically perform better/worse)? For Location (p11, L338-344): it is stated that locations of precipitation objects are not well represented, while the structure score seemed to suggest they were. Again, a bit more discussion is needed here. Agreed. We expanded the discussion of all three components. Specifically we made a distinction

Agreed. We expanded the discussion of all three components. Specifically we made a distinction between MC-type events and ARST-type events, in accordance also to reviewer #3 comments. The changes we made for the amplitude component are in L341-347 and L370-372; for the structure component are in L349-354 and for the location: L355-356, L360-366. Please note, we have tried to separate the discussion in the three different SAL components, and therefore do not address e.g., the intensity and the position when considering the structure component.

- The ellipticity of the 2d autocorrelation fields are discussed as a mean across all events. While in reality ellipticity would be expected to vary a lot between events (as indicated by the wide range of ellipticity values). Please justify why you think comparing means across events is useful?

A comparison of the median only is useful to understand whether the general orientation of the autocorrelation field is similar between observed and modelled rain. However, we also specifically presented the 10%-90% range. Both the median and the range are computed across all ( $\sim 10^4$ ) convective rain fields (i.e., n=11731 and 14323 for radar data and WRF data, respectively), and are not the inter-event medians (as in other parts of the manuscript). We did not want to add another figure to show this, but as can be seen in the figures below, the distribution of both the ellipticity values and its orientation is rather similar between the radar and the model.

Ellipticity (top panel) of the autocorrelation of convective rain fields from the radar (left) and the WRF model (n=11731 and 14323 for radar data and WRF data, respectively). The bottom panels show the orientation of the autocorrelation from the radar (left) and the WRF (right). Inner circles represent the number of rain fields having a similar orientation (in bins of 20 degrees).

- Discussion of performance differences: throughout section 4 (Results) you tend to attribute difference in performance between radar and WRF always to radar. You main observations at larger distance are subject to range degradation (L278, 296, 380), those at shorter distance are attributed to clutter (L372-373). In L397-398

you mention that "because radar QPE suffers from temporal inconsistencies". This way of discussing performance differences (basically attributing all deviations to radar) is unbalanced and I'm not convinced it's correct. Unless your radar product is particularly poor, but if that's the case, then what's left of the value of this "unique dataset"? Please check your performance evaluation in section 4 and make sure to have a more balanced discussion between the radar observations and WRF model results.

We thank the editor for pointing us to this bias in presenting the possible performance differences. We think that our radar dataset is unique not only because of its low value-bias (Marra and Morin, 2015), but also because of its long record. This long record enables to perform a quasiclimatological analysis. We also think that we showed in Sect. 4.2 that even though a correction procedure was applied for the radar QPE, it is not perfect (as seen in the comparison with rain gauges in Fig. 6). All that said, we do agree that we might have lessened the importance of model bias. Therefore, we added some statements regarding this possibility and changed other statements (L280-284, L300-306, L352-354, L360-364, L370-372, L404).

Discussion section:

- The first paragraph of the Discussion section has a lot of repetition of what's already reported in section 4. If it's meant to be a summary, it better fits at the end of section 4. Indeed it was meant to be a summary. We have moved it to Sect. 4.5.

- L445: "hither and tither" should be hither and thither. Still, it's an unusual term in professional manuscripts, so I suggest to rephrase it

Accepted and changed to "contrasting evidence" (L471).

- Section 5.2: a discussion on the usefulness of data in relation to flash flood comes out of the blue here and has no connection with anything in earlier parts of the manuscript. It's almost a lost piece of literature review. Consider to either move it up to Section 1 where literature is discussed or remove it entirely.

We have shorten this part of the discussion (L486-490). It is now more focused on our results, and less on previous studies. We, however, prefer to leave this part in, since it is one of the motivations to make such a study (e.g., L30-31: "Diverse rainfall patterns during HPEs cause different hydrological responses, thus an accurate representation of rainfall patterns during these events is crucial").

**- Section 5.3: please clarify whether 2nd paragraph (starting L484) still refers to ARST (as in first paragraph) or whether this is about all HPEs?**

It refers to all HPEs. We added "in general, for most HPEs" to clarify this point (L507).

**Conclusions:**

- You present a nice conclusion about the minimum scale that could be used for forecasting rainfall depths in relation to threshold intensities. It would be valuable nice to directly add some numbers here that came out for your analyses, to make the conclusion more quantitative.

We have changed this point in the conclusions a bit, separating it to two different points, with a more quantitative conclusion (L561-565).

**Figures:**

**- In Figure 7f a vertical scale seems to be missing, for the FSS scores?**

We thank the editor for noticing this problem. Indeed it was hidden behind the other panel, and we have brought it back to front.

- In Figure 11 vertical axis title seems incorrect: should be max accumulation instead of Rain threshold? Corrected.

Other changes we have made in this version compared to the previous one are in the affiliations of two of the authors, and in the supplementary (Table S1) – where one event was missing durations for which it was considered a HPE.

[revised manuscript text omitted]